# Correlation Analysis between Air Temperature and MODIS Land Surface Temperature and Prediction of Air Temperature Using TensorFlow Long Short-Term Memory for the Period of Occurrence of Cold and Heat Waves

**Jeehun Chung** [1] , **Yonggwan Lee** [1,]*, **Wonjin Jang** [1], **Siwoon Lee** [2] **and Seongjoon Kim** [3]

1 Department of Civil, Environmental and Plant Engineering, College of Engineering, Konkuk University, Seoul 05029, Korea; gop1519@konkuk.ac.kr (J.C.); jangwj0511@konkuk.ac.kr (W.J.)
2 DEEPNOID Inc., #1305, 55, Digital-ro 33-gil, Guro-gu, Seoul 08376, Korea; swlee@deepnoid.com
3 Division of Civil and Environmental Engineering, College of Engineering, Konkuk University, 120 Neungdong-ro, Gwangjin-gu, Seoul 05029, Korea; kimsj@konkuk.ac.kr
* Correspondence: leeyg@konkuk.ac.kr; Tel.: +82-2-444-0186

**Abstract:** The purpose of this study is to analyze the correlation between surface air temperature (SAT) and land surface temperature (LST) based on land use when heat and cold waves occur and to predict the distribution of SAT using the long short-term memory (LSTM) of TensorFlow. For the correlation analysis, the Terra and Aqua Moderate Resolution Imaging Spectroradiometer (MODIS) daytime and nighttime LST and maximum, minimum, and mean SAT were measured at 79 weather stations of the Korea Meteorological Administration (KMA) from 2008 to 2018. As a result of the correlation analysis between SAT and LST, the maximum SAT ($T_{MX}$) had a good correlation with the daytime LST of Terra MODIS, with a Pearson's correlation coefficient ($R$) of 0.92 and root mean square error (RMSE) of 4.8 °C, and the minimum SAT ($T_{MN}$) showed a good correlation with the nighttime LST of Terra MODIS, with an $R$ of 0.93 and RMSE of 4.2 °C. When analyzing temperature characteristics by land use (urban, paddy, upland crop, forest, grass, wetland, bare field, and water), it was confirmed that the climate mitigation effect of the wetland and vegetation area appeared in the LSTs and the observed SAT. In the cold wave period, the average temperatures for urban and wetland areas was the highest, and the average temperature for wetland and forest was not higher than that of other land use classes. As the SAT results predicted through the LSTM model, the accuracy of the $T_{MN}$ during the cold wave period was 0.59 for the coefficient of determination ($R^2$), 3.1 °C for RMSE, and 0.76 for the index of agreement (IoA), while the accuracy of the $T_{MX}$ for the heat wave period was 0.24 for $R^2$, 2.23 °C for RMSE, and 0.63 for IoA.

**Keywords:** air temperature prediction; heat and cold waves; long short-term memory; MODIS land surface temperature; TensorFlow

---

## 1. Introduction

In general, the occurrence of meteorological disasters increases the impacts and risks on the natural environment, society, and economy. Meteorological disasters are diverse, widespread, long-lived and dangerous. They are the largest disasters that cause people to lose their property [1]. Among them, heat waves are among the leading causes of environmentally related deaths worldwide and are expected to increase as a result of climate change. In South Korea, heat wave warnings are in effect when a maximum daily temperature of 33 °C or more is expected to last more than 2 days, and heat waves

are often accompanied by clear skies, atmospheric subsidence, suppressed air pollutant dispersion, and weak winds. During the heat wave period, heat and poor air quality are maintained due to high temperature, strong sunlight, and weak wind speed at the location, and they may stagnate and cause various diseases [2,3]. The main causes of heat waves include climate change—by increasing greenhouse gases in the atmosphere due to the use of fossil fuels and land use changes, such as deforestation and increased urban areas. On the other hand, cold wave warnings are issued when the minimum temperature in the morning is expected to be less than −12 °C for more than two days, and cold waves usually occur from late autumn until the spring of the following year. The main characteristics of cold waves are severe cold winds accompanied by snow, rain and frost, which have great adverse effects on agriculture, industry, transportation, and human health, causing great losses to the national economy [1].

For meteorological risk analyses, such as heat and cold waves over a wide range, the use of land surface temperature (LST) derived from thermal satellite sensors is essential because LST can modify the atmospheric boundary layer of the surface air temperature (SAT) in the surface energy balance [4]. Generally, ground observation data, such as SAT, can provide accurate weather information for the point in which the station is installed, but it is difficult to guarantee the reliability of the data for a wide range of spatial heterogeneity. Although data measured from satellites may have less accuracy than ground observations, it has the advantage of ensuring abundant time series data at a global scale. Additionally, these two types of temperature data can interact closely with each other and provide complementary information but are controlled by different physical mechanisms. LST is fundamentally altered by the Earth's surface energy balance, surface thermal properties, subsurface mediums, and atmospheric state, but SAT is primarily influenced by land cover, geographic location, weather conditions, etc. [5,6]. Therefore, it is necessary to verify the accuracy of both data types due to the insufficient temporal LST and spatial SAT for environmental and hydrological models.

Recently, LST time series data have been estimated for each optical satellite platform, such as Terra/Aqua Moderate Resolution Imaging Spectroradiometer (MODIS), Landsat, Geostationary Operational Environmental Satellite (GOES), and Advanced Very-High-Resolution Radiometer (AVHRR), and many studies have been conducted to analyze the relationship between SAT and LST through the temperature–vegetation index (TVX) method, energy balance models, and machine learning approaches. The key point of the TVX method is that the negative correlation between LST and the normalized difference vegetation index (NDVI) [7], along with the energy balance model, calculates the radiation balance with soil heat, latent fluxes, and sensible fluxes based on thermodynamics [8,9]. Statistical techniques, including simple and multiple linear regression models, are a method of estimating SAT using the correlation between SAT and other variables, such as LST, NDVI, solar radiation, and elevation [10–13]. Last, machine learning has been actively researched recently, such as neural networks, random forest, and cubist methods, and machine learning has proven its effectiveness in estimating SAT from LST with additional variables [14–18]. Among them, deep learning is a special part of machine learning algorithms based on a multilayer neural network (MLP-NN) with multiple layers of nonlinear information [19], and it has strengths in dealing with noise, missing data, and outliers [20]. In particular, recurrent neural networks (RNNs) introduce the concept of sequence design of the algorithm, which shows better performance for time series analysis [21]. However, when the traditional RNN models are applied to long temporal data, they could create vanishing gradient and exploding gradient problems [22]. Long short-term memory (LSTM) networks that have special hidden units have been proposed for overcoming this difficulty and for storing long time series data [23], and they have been applied to remote sensing data, such as classification hyperspectral data, land cover change detection, and crop yield prediction [24–26]. Several studies have been conducted using LSTM to calculate accurate LST [27,28] or to analyze the impact on global warming through the trend analysis of LST [29]. However, few studies have been used to analyze the correlation between SAT and LST using LSTM.

There are two main purposes for conducting this study: the first purpose is to analyze the characteristics of LST and SAT based on land use during heat and cold waves, and the second purpose is to predict the spatial distribution of SAT from LST by applying the LSTM model. For the correlation analysis, ground measured SAT and Terra and Aqua MODIS LSTs were prepared with land use (Figure 1a), and the characteristics of SAT and LST were identified by land use classes (Figure 1b). Based on the correlation and characteristic analysis results, an SAT prediction model for the heat wave and cold wave periods was developed for each land use class by using TensorFlow-LSTM (Figure 1c). The model results were verified with the observed SAT.

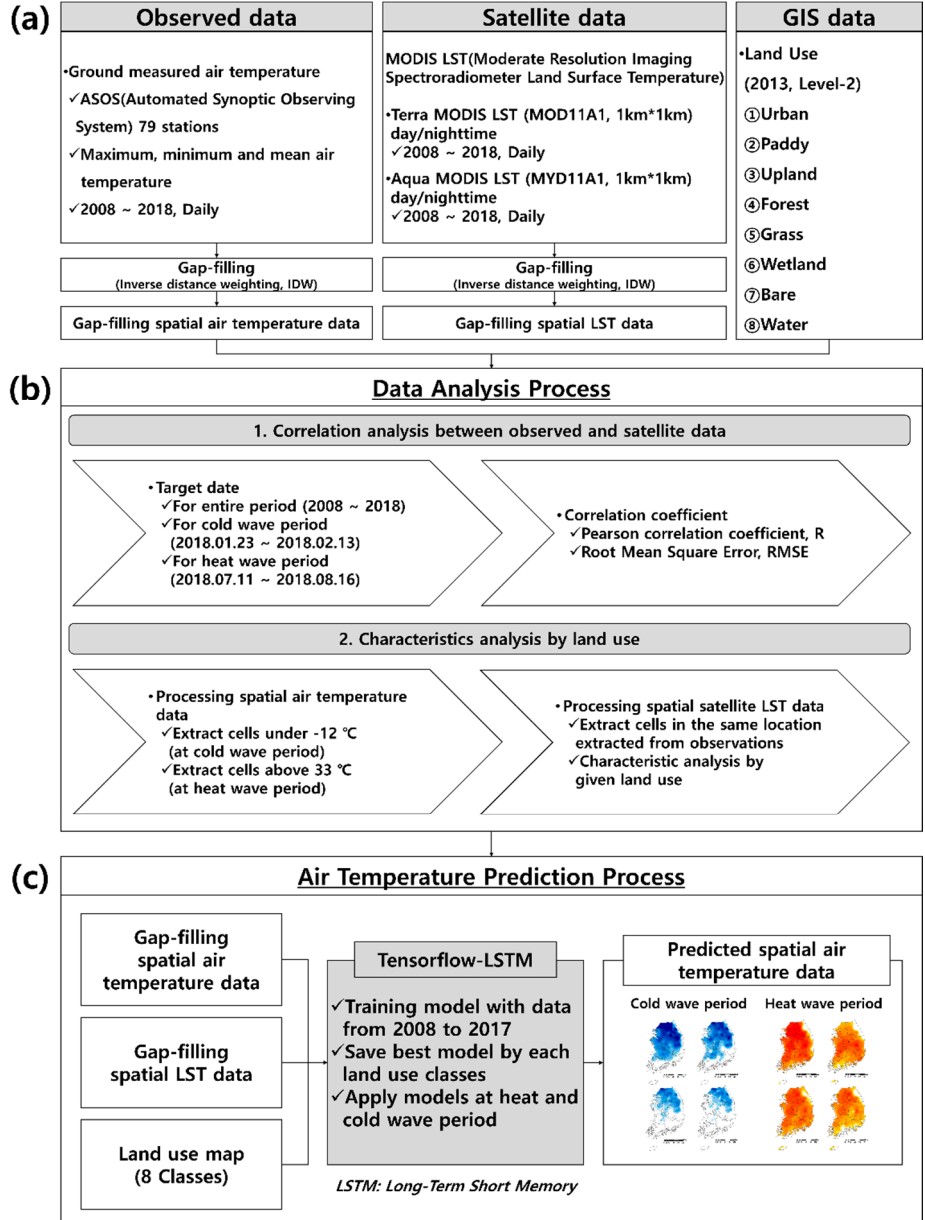

**Figure 1.** Flowchart of the study: (**a**) a brief description of datasets, (**b**) data analysis (correlation and characteristics) process, and (**c**) air temperature prediction.

## 2. Materials and Methods

### 2.1. Study Area

The study includes is all regions of South Korea (124°5'E to 130°0'E and 33°8'N to 39°0'N), located in northeast Asia. The Korea Meteorological Administration (KMA) monitors weather conditions (temperature, wind speed, precipitation, etc.) automatically using the 494 Automatic Weather System (AWS) and 96 Automated Synoptic Observing System (ASOS) [30]. ASOS is installed in local government offices, branch offices, weather stations, and observation stations to observe weather phenomena and data sharing using international expertise. AWSs are installed in places that are difficult for humans to access, such as in mountainous areas or islands, and real-time, dangerous weather events, such as torrential rain, hail, thunderstorms, and gusts (https://www.kma.go.kr), are monitored. The AWS observations by the KMA began in 1988 and has increased to over 200 points since 2000. However, the stabilization verification of data quality is insufficient, and AWSs are more likely to be contaminated than ASOS due to more frequent branch movements [31]. Therefore, for the observed maximum, minimum, and mean SAT ($T_{MX}$, $T_{MN}$, and $T_{MM}$) data used in this study, 79 stations of total ASOS station data were used daily from 2008 to 2018, except for the stations with no data for the period or out of the range of land use map and AWSs (Figure 2a).

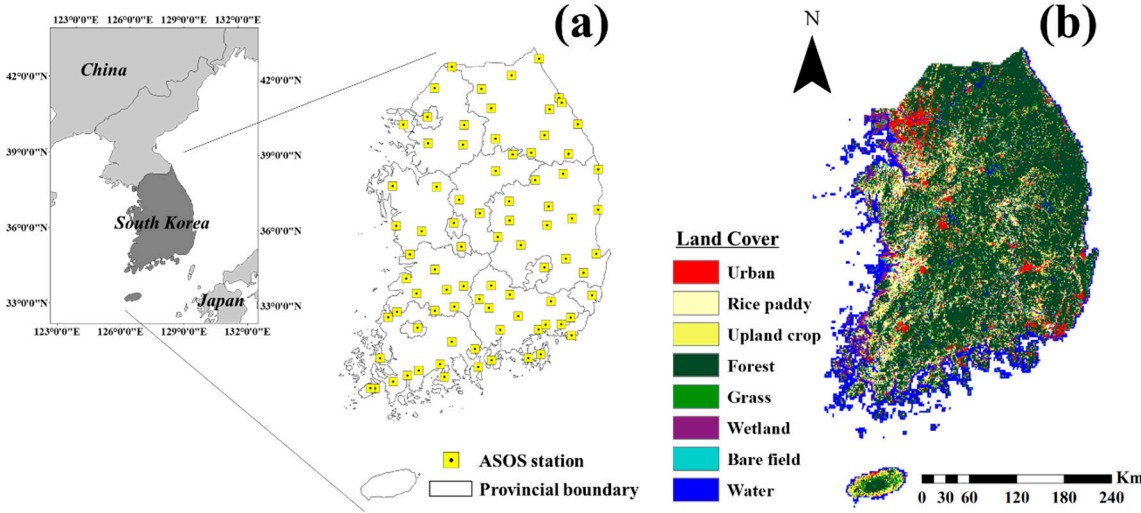

**Figure 2.** Relevant information on the study area. (**a**) Location of 79 Automated Synoptic Observing System (ASOS) weather stations and (**b**) Resampled land use map.

In South Korea, with a land area of 99,000 km$^2$, more than 65% of the total area is mountainous, and 12.2% of the lowland is composed of agricultural areas. The land use map was resampled from the level 2 land cover map provided by the Ministry of Environment (MOE) with a 1000 m spatial resolution, which is the same as the MODIS LST product. Cultivated areas are subdivided into rice paddy and upland crops with different soil characteristics. Finally, the resampled land cover map was reclassified into 8 categories: urban, rice paddy, upland crop, forest, grass, wetland, bare field, and water (Figure 2b). Based on the reclassified land cover map, the sum of the total area and the average elevation and latitude of each land cover type are shown in Table 1. Forest, water, rice paddy and upland crops accounted for the largest area, on the order of 55.4%, 11.3%, 9.0% and 8.0%, respectively. In the case of water, the area was larger than that announced by the MOE, which is considered to reflect the land cover map both inland and in coastal areas. The altitudes were high in upland crops, forests, grasses, and bare fields, and the latitudes showed little difference by land use.

**Table 1.** Area, elevation, and latitude of resampled land cover.

| Land Use | Area (km², %) | Elevation (m) | Latitude |
|---|---|---|---|
| Urban | 5589 (5.1) | 77.0 | 36.3 |
| Rice paddy | 9877 (9.0) | 74.1 | 36.0 |
| Upland crop | 8751 (8.0) | 145.9 | 36.0 |
| Forest | 60,490 (55.4) | 333.7 | 36.4 |
| Grass | 6800 (6.2) | 173.2 | 36.1 |
| Wetland | 3190 (2.9) | 29.5 | 35.9 |
| Bare | 2166 (2.0) | 141.1 | 36.3 |
| Water | 12,295 (11.3) | 15.7 | 35.5 |
| Total | 109,158 | - | - |

The climate in South Korea is divided and characterized into four seasons (see Appendix A), with a wet monsoon summer season called Jangma from June to August. In the Jangma season, precipitation is concentrated, which accounts for approximately 50–60% of the annual precipitation of 1300 mm, and typhoons often occur between summer and autumn. In spring and autumn, relatively dry climates persist under the influence of migratory anticyclones. In winter, it is relatively cold compared with other regions at the same latitude because of the influence of the Siberian anticyclone [32].

The average annual temperature in South Korea is between 10 and 15 °C, with January being the coldest at −6 to 3 °C and hottest at 23 to 26 °C in August. During the last 106 years (1912–2017), the mean annual temperature in South Korea was 13.2 °C, and in the 2010s (2011–2017), it was 14.1 °C, indicating recent warming. In recent decades, the extreme climate phenomena associated with low temperatures have increased, showing a trend that is different from the trends of the last 30 years, but the seasonal duration of summer has been lengthened by 19 days from 98 to 117 days, and winter has been shortened by 18 days from 109 to 91 days [33].

### 2.2. Terra/Aqua MODIS LST

The Moderate Resolution Imaging Spectroradiometer (MODIS) is the key instrument aboard the Terra and Aqua satellites launched in December 1999 and May 2002, respectively, by the National Aeronautics and Space Administration (NASA). Both the Terra and Aqua platforms were launched into a 705 km sun-synchronous near-polar orbit with a 2330 km swath and ±55-degree scanning pattern to scan the entire earth's surface every 1 to 2 days and acquire data with high radiometric sensitivity (12 bit) in 36 spectral bands ranging from 0.4 μm to 14.4 μm (https://modis.gsfc.nasa.gov/). Terra's equatorial crossing local time in a descending (ascending) orbit is 10:30 AM (10:30 PM). In contrast, Aqua crosses the equator at 1:30 PM (1:30 AM) local time in descending (ascending) orbit.

The MODIS LST daily products are constructed by using the generalized split window algorithm [34,35], which uses the 31 and 32 bands of MODIS spectral bands. The level 3 LST product (MOD11A1 for Terra and MYD11A1 for Aqua) is produced in tiles of 1200 rows by 1200 columns at an approximate spatial resolution of 1 km [36]. From these daily LST data, daytime and nighttime LSTs for each Terra and Aqua platform ($LST_{TD}$, $LST_{TN}$, $LST_{AD}$, and $LST_{AN}$) were prepared from 2008 to 2017 and cold wave and heat wave period in 2018 from EARTHDATA (https://earthdata.nasa.gov/). Thus, 3712 $LST_{TD}$, $LST_{TN}$, $LST_{AD}$, and $LST_{AN}$ images, 14848 in total, were utilized for the analysis and prediction process.

Before application to the analysis, only pixels of the highest quality, i.e., cloud free and error free pixels, can be used. For that reason, quality flag (QF) and quality control (QC) data from each LST data were extracted and examined manually to exclude cloud-contaminated pixels with unreasonably low LST values. Quality-controlled LST data (digital number: DN) were converted to degrees Celsius by the following equation:

$$C = 0.02 * DN - 273.15 \tag{1}$$

where *C* is the Celsius temperature and 0.02 is the scale factor of the MODIS LST data and DN (Digital number) is the value of the raw LST data.

## 2.3. TensorFlow-LSTM

In this study, predicting the SAT at a time t based on an LST time-series dataset at a time t-1 at a point location of each weather station was performed with LSTM, an advanced form of RNN, one of the deep learning algorithms. A RNN is a class of artificial neural networks (ANNs) that have been adopted to surmount the challenge of complicated long-range persistence in traditional approaches, such as feedforward neural networks (FNNs) and time-delay neural networks (TDNNs) [37,38]. The RNN uses the temporal relationship among the inputs in the training step with intrinsic dynamic memory provision from the constituent in the recurrent linking, and its structure can flexibly vary according to the user's needs [37]. However, the major problem of RNNs is that they are severely affected by the extinction gradient problem, which may unlimitedly increase, leading to network disruption [22]. Hochreiter and Schmidhuber [23] suggested a new concept of RNN, known as LSTM, to avoid this problem.

Figure 3 shows the basic LSTM architecture. As shown in Figure 3, the main advantage of the LSTM is that traditional neuron units replaced with LSTM in the hidden layer of RNN with a memory block consisting of forget, input, and output gates. These gates, consisting of a layer of sigmoid and pointwise multiplication, control the data through the cell and neural networks. The advantage of a recurrent neural network is that it can reflect previous information to the current information. However, to solve the long-term dependence problem, it is necessary to remember not only previous short-term memory but also long-term memory [39]. Therefore, in LSTM, information is stored and used in a separate Cell ($C_t$). Additionally, since the information stored in $C_t$ must be forgotten if it is not related to the current information, the $C_t$, and the information to be output ($h_t$) are separately calculated. Here, the input and forget gate are judges the information to remember or whether forget. In LSTM, through this forget gate, it is possible to prevent the problem that the computational amount may explode using all the past information in the RNN by cutting unnecessary information. Thus, LSTM is suitable for effectively mitigating the vanishing gradient problem and for dealing with complex problems with long-term dependency [28]. The LSTM equations used in this study are detailed in Appendix B.

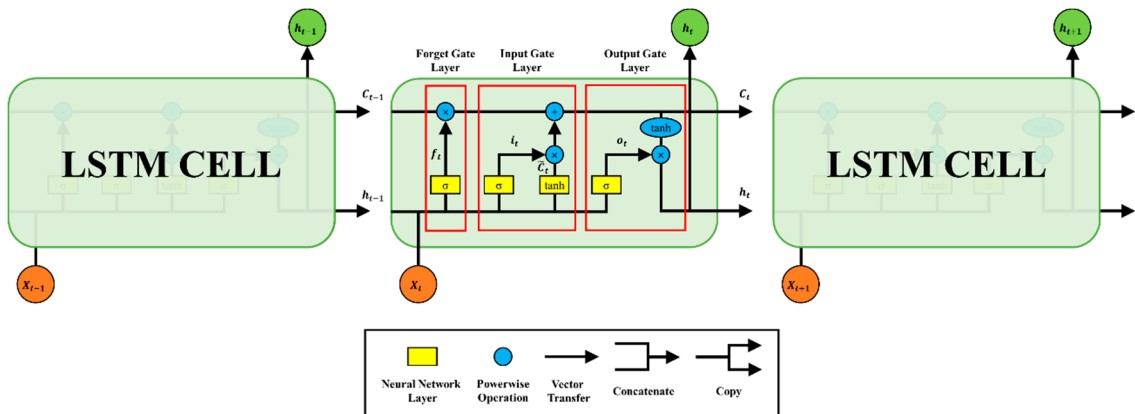

**Figure 3.** Basic LSTM Architecture.

TensorFlow is the open source machine learning software library developed by Google Brain in November 2015. Google Brain was launched in 2011 by a team of Google researchers and Professor Andrew Ng of Stanford University. Their goal was to create a large-scale deep learning software system; as a result, DistBelief has appeared. DistBelief is a large-scale distributed machine learning system, a cloud service that runs on the Google Cloud platform. While DistBelief was Google's first-generation machine learning infrastructure, TensorFlow is a second-generation machine [40].

TensorFlow includes more than 480 developers to improve software quality, and wrapper libraries, such as TFLearn, TensorLayer, and PrettyTensor, have appeared. One year after its release, TensorFlow became one of the most popular machine learning libraries [41,42].

The advantage of using TensorFlow in modeling LSTM is its simplicity of installation and its ease of customization using Python language, also in tracking the operation and backpropagation properties is unnecessary. TensorFlow replaces the tracking process and automatically updates model variables with gradients calculated based on cost functions, optimization functions, and learning rates [43]. Additionally, TensorFlow-GPU, which is a graphic processing unit (GPU) accelerated version of TensorFlow, was adopted to improve the execution speed of the LSTM model.

## 3. Results

### 3.1. Analysis of Correlation between MODIS LST and SAT with Descriptive Statistics

Descriptive statistics were performed to analyze the correlation between MODIS LST ($LST_{TD}$, $LST_{TN}$, $LST_{AD}$, and $LST_{AN}$) and SAT ($T_{MX}$, $T_{MM}$ and $T_{MN}$) on each dataset from 2008 to 2018. Overall, all data showed a negatively skewed distribution with a median value larger than the average value. Among the averages of the four LST datasets, the $LST_{AD}$ was the highest at 17.9 °C, which was closest to 18.4 °C, and the average of the $T_{MX}$ and the $LST_{TN}$ was the lowest at 7.4 °C. In the case of standard deviation (SD), $LST_{TD}$ and $LST_{AD}$ of daytime data were 10.5 and 10.4, respectively, showing a large scatter, and $LST_{TN}$ and $LST_{AN}$ of nighttime data were 7.4 and 6.3, respectively. For all the data, the mean value was the highest for the $T_{MX}$, and the range of observations was the largest in $LST_{AD}$ from −20.6 °C to 42.1 °C (Figure 4).

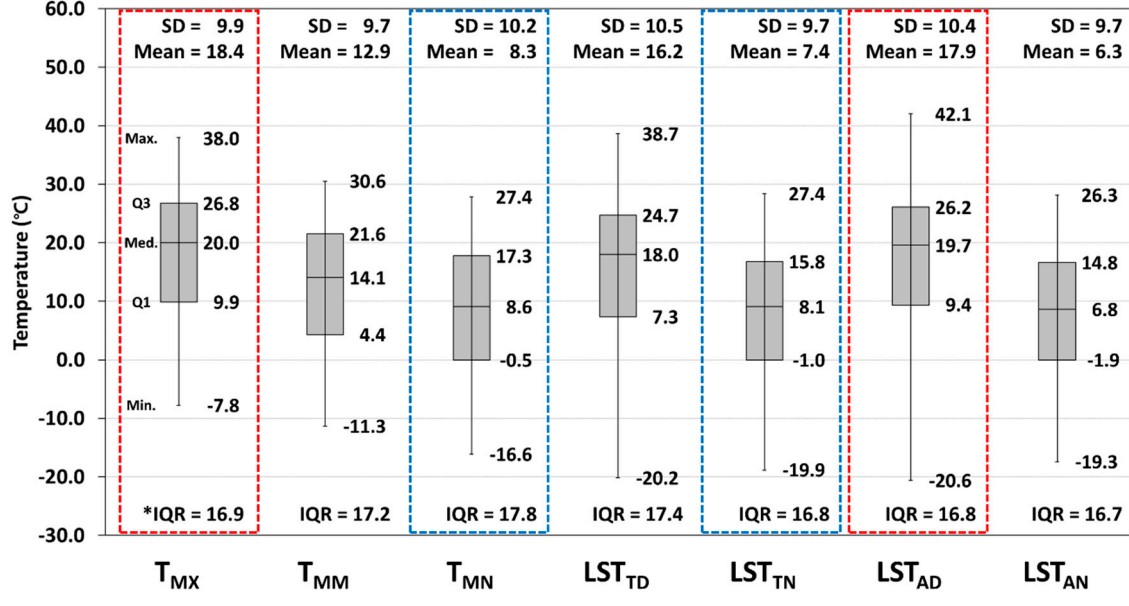

**Figure 4.** Boxplots of observed maximum, minimum, and mean SAT ($T_{MX}$, $T_{MM}$, $T_{MN}$, respectively) and four land surface temperatures (LSTs) ($LST_{TD}$, $LST_{TN}$, $LST_{AD}$, $LST_{AN}$) for 79 ASOS stations from 2008 to 2018. Among the four LST datasets, the average for $LST_{TD}$ was the highest, similar to the $T_{MX}$, and the $LST_{TN}$ average was the closest to the $T_{MN}$. IQR: interquartile range.

Between MODIS LST and SAT, the correlation was analyzed using root mean square error (RMSE) and Pearson's correlation coefficient (*R*) (Figure 5 and Table 2). The Pearson's correlation coefficient shows values in the range of −1 to 1, and values closer to 1 or −1 indicate that there is a positive or negative correlation, respectively, and the closer to zero the value is, the lower the correlation.

RMSE means that the closer to 0 the value is, the better the correlation [44]. Figure 5 compares the time series distributions of the SAT (black solid line) and the Terra MODIS LSTs (hollow red dot) and Aqua MODIS LSTs (hollow blue dot) at the weather stations.

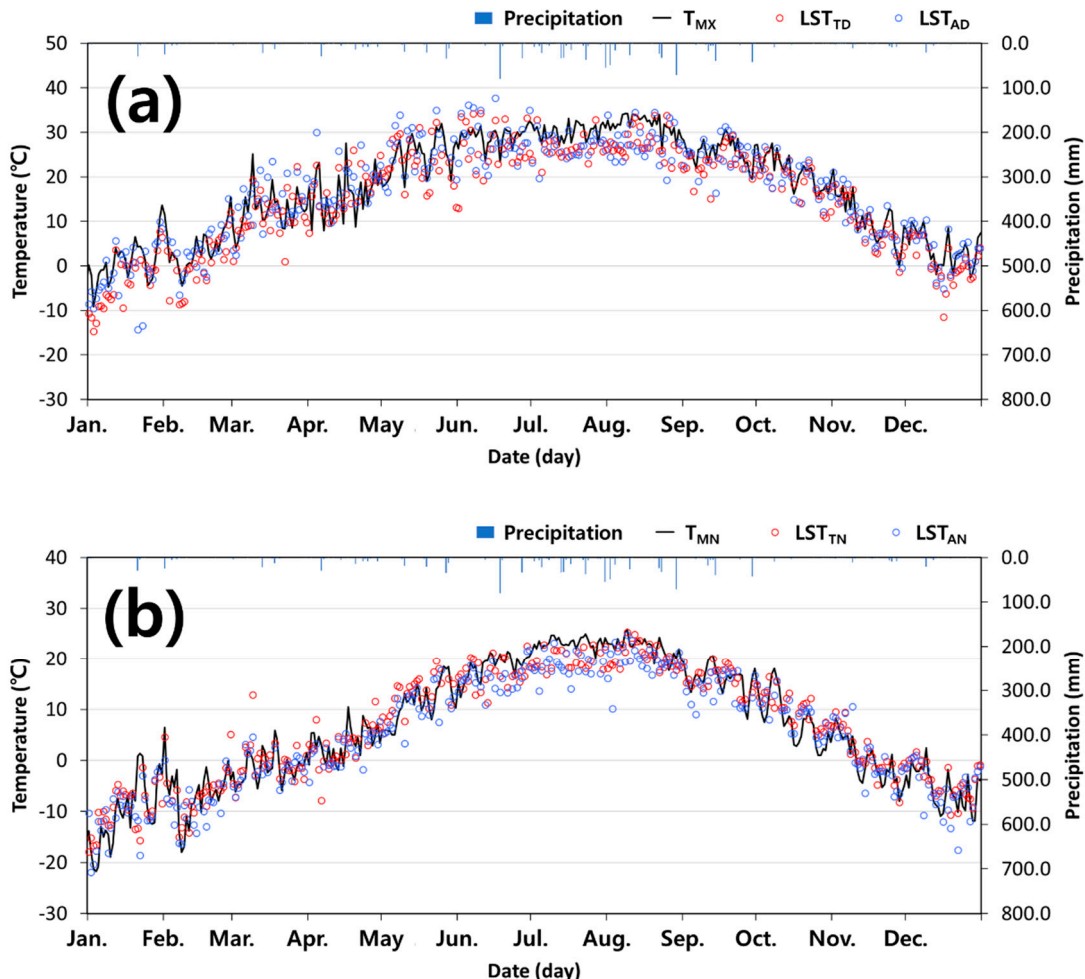

**Figure 5.** Comparison between (**a**) daytime MODIS LSTs (LST$_{TD}$ and LST$_{AD}$) and T$_{MX}$, (**b**) nighttime MODIS LSTs (LST$_{TN}$ and LST$_{AN}$) and T$_{MN}$ for the Chungju weather station of the average year.

**Table 2.** Comparison of SATs (T$_{MX}$, T$_{MM}$, and T$_{MN}$) and LSTs (LST$_{TD}$, LST$_{TN}$, LST$_{AD}$, and LST$_{AN}$) for the whole period, cold wave period, and heat wave period.

| Index | Data Type | Whole Period (2008~2018) | | | | Cold Wave Period (2018.01.23~2018.02.13) | | | | Heat Wave Period (2018.07.11~2018.08.16) | | | |
|---|---|---|---|---|---|---|---|---|---|---|---|---|---|
| | | LST$_{TD}$ | LST$_{TN}$ | LST$_{AD}$ | LST$_{AN}$ | LST$_{TD}$ | LST$_{TN}$ | LST$_{AD}$ | LST$_{AN}$ | LST$_{TD}$ | LST$_{TN}$ | LST$_{AD}$ | LST$_{AN}$ |
| R | T$_{MX}$ | 0.92 | 0.94 | 0.90 | 0.94 | 0.73 | 0.73 | 0.72 | 0.78 | 0.36 | 0.39 | 0.37 | 0.42 |
| | T$_{MM}$ | 0.90 | 0.95 | 0.87 | 0.95 | 0.70 | 0.75 | 0.66 | 0.79 | 0.32 | 0.39 | 0.31 | 0.43 |
| | T$_{MN}$ | 0.86 | 0.93 | 0.82 | 0.93 | 0.53 | 0.60 | 0.48 | 0.59 | 0.20 | 0.18 | 0.20 | 0.24 |
| RMSE | T$_{MX}$ | 4.8 | 11.3 | 4.8 | 12.3 | 3.4 | 10.1 | 3.6 | 10.9 | 5.8 | 11.9 | 5.4 | 12.5 |
| | T$_{MM}$ | 5.9 | 6.0 | 7.5 | 7.0 | 5.2 | 5.2 | 7.7 | 5.7 | 4.4 | 6.5 | 5.3 | 7.3 |
| | T$_{MN}$ | 10.2 | 4.2 | 11.9 | 4.3 | 10.1 | 4.7 | 13.4 | 4.7 | 6.9 | 3.8 | 7.9 | 4.2 |

LST$_{TD}$: Terra MODIS LST daytime, LST$_{TN}$: Terra MODIS LST nighttime, LST$_{AD}$: Aqua MODIS LST daytime, LST$_{AN}$: Aqua MODIS LST nighttime, T$_{MX}$: Maximum SAT, T$_{MM}$: Mean SAT, and T$_{MN}$: Minimum SAT, RMSE: root mean square error.

Figure 5a compare daytime LSTs and T$_{MX}$, and Figure 5b compare nighttime LSTs and T$_{MN}$, respectively, for average year. The overall variation in all MODIS LSTs was similar to the observed

SAT ($T_{MX}$, and $T_{MN}$), but the nighttime LSTs showed less visual variability, and these trends could be confirmed by the interquartile range (IQR). The IQR of $LST_{TD}$ was 17.4, which was larger than that of $LST_{TN}$ by 0.6, and $LST_{AD}$ was 16.8, which was slightly higher than $LST_{AN}$ by 0.1. Concerning $T_{MX}$ (Table 2), the $R$ of $LST_{TD}$ and $LST_{AD}$ showed values of 0.94, and 0.93, respectively. In the case of RMSE, the daytime LSTs ($LST_{TD}$ and $LST_{AD}$) were calculated to be 4.7 °C and 4.6 °C, respectively. In $T_{MN}$ (Table 2), the RMSEs of $LST_{TN}$ and $LST_{AN}$ were 4.0 °C and 4.1 °C, respectively, showing a smaller difference from the nighttime LSTs. In other words, the time series change pattern showed that the nighttime LSTs were better correlated with each of the SAT data than the daytime LSTs, but the daytime LSTs showed a small deviation from $T_{MX}$, and in the same vein, the nighttime LSTs showed a small deviation from the $T_{MN}$. Since the $R$ of the time series variation analysis is highly correlated with more than 0.93 in all LST data, $T_{MX}$ should use $LST_{TD}$ with a small deviation, and $T_{MN}$ should use $LST_{TN}$ to estimate the spatial distribution of each temperature data using TensorFlow-LSTM.

According to the KMA press release [45], a strong cold wave occurred for 22 days from January 23 to February 13, 2018. At that time, the nation's highest SAT was 0.6 °C, and the daytime SATs remained in the subzero region. On January 27, Gunsan recorded the lowest SAT of −15.0 °C, and Sancheong was −14.6 °C, and on February 7, Jinju recorded −14.3 °C, which was the lowest minimum SAT since 1973. The national average heat wave duration in 2018 was 29.2 days, which was very different from the 8.6-day average, and it was the longest heat wave since 1973. For the duration of heat waves by region, Geumsan lasted for the longest period at 37 days from July 11 to August 16, followed by Gwangju at 36 days from July 12 to August 16. Therefore, this study set the cold wave period between January 23 and February 13, 2018 and the heat wave period between July 11 and August 16 and analyzed the correlation between MODIS LST and SAT for each period.

In the cold wave period (Table 2), $LST_{AN}$ was most correlated for all SAT data ($T_{MX}$, $T_{MM}$, and $T_{MN}$). The $R$ values of $T_{MX}$ and $T_{MM}$ for $LST_{AN}$ were 0.78 and 0.79, respectively, and the $R$ of $T_{MN}$ was 0.59, which was lower than that of $LST_{TN}$, but it was higher than that of other LST data. Similar to the previous time series analysis, the nighttime LSTs showed a higher correlation with SAT, and in particular, the $R$ of $T_{MM}$ was the highest at 0.75 ($LST_{TN}$) and 0.79 ($LST_{AN}$). The correlation with the $T_{MN}$ was expected to be the highest during the cold wave period, but the correlation coefficients for all LST data were somewhat inferior to the mean and $T_{MX}$, which is because of the difference in the measurement times of the satellites. Usually, the $T_{MN}$ of the day occurs just before sunrise, but the measurement times of the Terra and Aqua MODIS were 10:30 (AM and PM) and 1:30 (AM and PM), respectively. Overall, the RMSE showed better results in the cold wave period than in the whole period. On the other hand, the heat wave period was analyzed and lacked correlations among SAT data and Terra/Aqua MODIS LSTs. The $R$ for $T_{MX}$ showed values of 0.36, 0.39, 0.37, and 0.42 for $LST_{TD}$, $LST_{TN}$, $LST_{AD}$, and $LST_{AN}$, respectively, and RMSE was 5.8 °C, 11.9 °C, 5.4 °C, and 12.5 °C, respectively. The RMSE of the daytime LSTs ($LST_{TD}$ and $LST_{AD}$) increased by 1.0 °C and 0.6 °C, respectively, and $R$ showed significantly worse results.

### 3.2. Characteristics Analysis by Land Use during Heat and Cold Waves

Time series analysis of the $T_{MN}$ and $T_{MX}$ was performed on 79 ASOS stations for the heat and cold wave periods (Figure 6). In cold wave period, strong air pressure in the upper part of the Ural Mountains and the Bering Sea caused the east–west flow of the atmosphere to stagnate, and cold air in the vicinity of Siberia continued to flow in and out of South Korea, which is located between the upper part of the Bering Sea [45]. On January 30 and February 7, when a strong cold wave occurred, the temperature dropped sharply at most stations compared with the other days, but on January 27, some stations showed an increase in $T_{MN}$. It seems that there is a difference in the time of occurrence of cold waves by region and latitude. In the case of the Gunsan and Sancheong stations, January 27 showed the lowest minimum temperature.

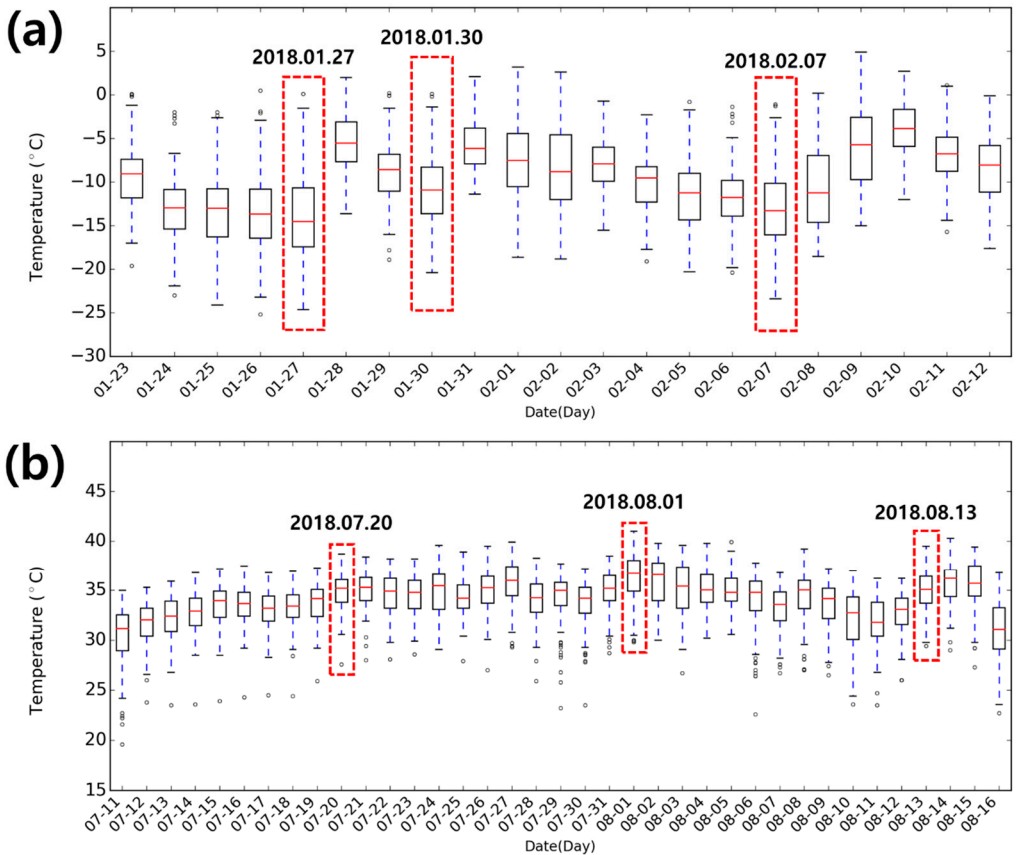

**Figure 6.** Time series boxplots of (**a**) $T_{MN}$ of 79 ASOS stations during the cold wave period from January 23 to February 13 and (**b**) $T_{MX}$ of 79 ASOS stations during the heat wave period from July 11 to August 16 in 2018 [46].

In early July, Tibetan and North Pacific anticyclone developed unusually strongly and remained until August, with hot air flowing in continuously. In addition, heated water vapor flowed into the Korean Peninsula due to strong solar radiation, and the east wind effect was added to the typhoon's frequent north effect, so the hot weather continued [45]. During the heat wave, most of the stations showed a sharp rise in $T_{MX}$ on July 20, August 1, and August 13. In particular, Baeknyeongdo and Taebaek were found to have many days when the daily $T_{MX}$ in the heat wave period was below 30 °C. As with the cold wave period, it is judged that there is a difference depending on the observed elevation and latitude, and in the case of Baeknyeongdo, the temperature reduction effect was shown by the topographical characteristics of the island region. In this study, we analyzed the characteristics between SAT and LSTs according to the land use that showed a sharp change in time series analysis for each of the three days. At this time, the characteristics of the cold wave period were analyzed using the $T_{MN}$ and $LST_{TN}$, and during the heat wave period, the characteristics were analyzed using the $T_{MX}$ and $LST_{AD}$.

As a result of analyzing the characteristics of each land use for the cold wave period, it was confirmed that the climate mitigation effect of the wetland and vegetation area appears in the LSTs and the SAT observation data (Figure 7). In Figure 7, the solid red line represents −12 °C, which is the reference temperature of the cold wave. For $T_{MN}$ on January 27 (Figure 7a), the forest average was the lowest at −16.9 °C, the bare field was −16.7 °C, the water was −16.6 °C, and the upland crop and grass were both −16.5 °C. The average temperatures of the wetland and urban areas were −16.2 °C and −16.3 °C, respectively. In urban areas, the average temperature was higher than that of the other land uses because of the heat island effects caused by dense buildings, and wetlands were considered

to have a high average temperature because of climate mitigating effects. In the case of rice paddies, the average was the highest at −15.8 °C, and it was judged to be influenced by the artificial heat generated from the plant cultivation site, such as greenhouses [47]. In the winter in South Korea, winter crops, such as barley, wheat, garlic, feed crops, etc., are grown in rice paddy fields or potatoes are grown in greenhouses. The minimum value was the lowest in the forest at −24.6 °C, and the wetland was −22.4 °C, which is a 2.2 °C difference from the forest, indicating the mitigation effect of wetlands. In LST$_{TN}$, forests had the highest average of −12.7 °C, followed by upland crops and wetlands at −12.9 °C and −13.0 °C, respectively.

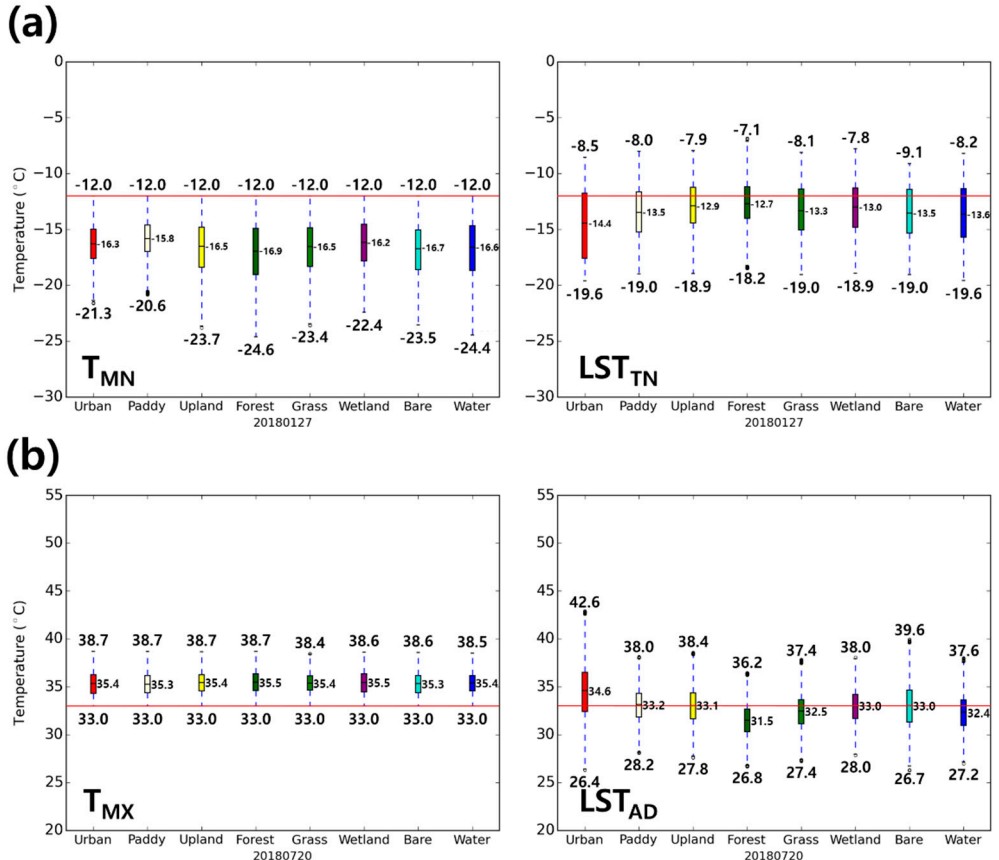

**Figure 7.** Boxplots of (**a**) observed T$_{MN}$ and LST$_{TN}$ for 8 land use classes (red: urban, light yellow: paddy, yellow: upland crop, dark green: forest, green: grass, purple: wetland, cyan: bare field, and dark blue: water) on January 27, 2018, and (**b**) observed T$_{MX}$ and LST$_{AD}$ on July 20, 2018. In (**a**), the red solid line on the graph is -12 °C, which represents the reference temperature of the cold wave and in (**b**), the red solid line on the graph is 33 °C, which represents the reference temperature of the heat wave.

The SAT and LST characteristics of the heat wave period were similar to those of Figure 7a (Figure 7b). The T$_{MX}$ was always reasonably above the heat wave reference temperature, while LST$_{AD}$ was above or below the reference temperature. In addition, T$_{MX}$ did not show a significant difference in average temperature by land use, whereas LST$_{AD}$ showed a significant difference. The average temperature of the forest and wetland was 35.5 °C for T$_{MX}$ on July 20 (Figure 7a), and the average temperature of the other land uses did not differ significantly at 35.3 °C to 35. 4 °C. In LST$_{AD}$, the average temperature of urban areas was the highest at 34.6°C, and the difference from wetlands was 1.6°C. The bare field was 33.0 °C, the same as the wetland, but the maximum value was 39.6 °C, which was the second highest after the urban area. This finding was because vegetation cover was scarce, so it was not affected by the climate mitigation effects. On the other hand, the maximum values of paddy, upland crop, and grass were 38.0 °C, 38.4 °C, and 37.4 °C, respectively, which were lower

than those of urban and bare field areas. The forest had the lowest temperature of 31.5 °C by regulating the amount of evapotranspiration [48]. The average temperature of the water was 32.4 °C, which was the second lowest after the forest due to the large heat capacity. The box plots for the other days of heat and cold wave can be found in Appendix C.

### 3.3. SAT Prediction Using TensorFlow-LSTM

Based on the results of 3.1 and 3.2, SAT prediction ($T_{MX}$ and $T_{MN}$) considering land use characteristics using TensorFlow-LSTM was performed during heat and cold waves in 2018. As input data, four MODIS LST datasets ($LST_{TD}$, $LST_{TN}$, $LST_{AD}$, and $LST_{AN}$) were used daily from 2008 to 2017. The prediction model was prepared for each of the eight land use classes (urban, paddy, upland crop, forest, grass, wetland, bare field, and water), and each model was composed of single-layered LSTM with 100 hidden layers. The sequence length was set to one to predict the SAT after a day. The mean squared error (MSE) was used as a loss function, and the Adam optimizer was selected to minimize the losses. The most important factor when using machine learning was the number of epochs. Too many epochs caused overfitting, and too few epochs caused underfitting. To solve overfitting and underfitting problems, the early stopping function in TensorFlow was used. The early stopping function stopped training when a monitored quantity stopped improving. In this study, the maximum epoch was set to 300, and when there was no increase in performance over one-fifth of maximum epoch (60 epochs), the training process was terminated, and the model was saved as the best model. The trained model was applied for the cold and heat wave periods in 2018 for verification.

Table 3 shows the statistical indexes of the prediction results for the $T_{MN}$ of the cold wave period and $T_{MX}$ of the heat wave period for eight land use classes. The coefficient of determination ($R^2$), index of agreement (IoA), and RMSE were used as the objective functions to determine the correlation between the simulation results and the observed data. The $R^2$ and IoA means that the closer to one the value is, the higher the correlation between the simulated data and observed data. All statistical results of each 79 meteorological station are detailed in Appendix D. Overall, the climate mitigation effect of vegetation and water appeared in the LSTM model results during the heat and cold wave periods. In the vegetation area during the cold wave period, i.e., rice paddy, upland crop, forest, grass, and water, $R^2$ was 0.61, 0.66, 0.59, 0.60, and 0.65, respectively, which were higher than those of non-vegetation areas, such as urban and bare fields. In the wetland, $R^2$ was the lowest at 0.5, while the IoA was the highest at 0.82, and the RMSE also showed the smallest deviation of 2.09 °C. In contrast, during the heat wave period, the $R^2$ was higher in urban and bare fields than in vegetation areas at 0.27 and 0.32, respectively. In the water and upland crops, $R^2$ was the lowest at 0.11 and 0.18, respectively, and IoA also had the lowest at 0.51 in the water. The reason for this tendency was judged to be that cold waves tended to appear continuously for several days after occurring in an area under the influence of the Siberian anticyclone, while the frequency of heat waves was uncertain for various reasons, such as the rise in average temperature due to global warming and the heat island effect due to urbanization. In addition, in the case of vegetation and water areas, due to the high heat capacity, the temperature change in the region was less than that of urban and bare fields. In this study, the SAT predicted by LSTM was suitable for reproducing the trend of continuous temperature change in vegetation and water areas during the cold wave period, but it seemed that the prediction was weak for the uncertain change in the heat wave period. Another reason was that the quality of LST data was poor due to the influence of clouds in summer.

**Table 3.** The statistical indicator ($R^2$, RMSE, and IoA) results of 8 land use classes during the cold wave period (2018.01.23.–2018.02.13.) and heat wave period (2018.07.11.–2018.08.16.).

| Land Use [a] | Cold Wave Period (2018.01.23.–2018.02.13.) | | | Heat Wave Period (2018.07.11.–2018.08.16.) | | |
|---|---|---|---|---|---|---|
| | $R^2$ | RMSE (°C) | IoA | $R^2$ | RMSE (°C) | IoA |
| Urban (37) | 0.54 | 4.12 | 0.67 | 0.27 | 2.13 | 0.66 |
| Rice paddy (9) | 0.61 | 3.18 | 0.74 | 0.21 | 1.99 | 0.60 |
| Upland crop (12) | 0.66 | 3.08 | 0.78 | 0.18 | 2.12 | 0.61 |
| Forest (12) | 0.59 | 2.65 | 0.80 | 0.26 | 2.24 | 0.64 |
| Grass (7) | 0.60 | 3.85 | 0.67 | 0.23 | 2.35 | 0.59 |
| Wetland (1) | 0.50 | 2.09 | 0.82 | 0.35 | 2.79 | 0.69 |
| Bare field (2) | 0.54 | 2.25 | 0.82 | 0.32 | 2.08 | 0.71 |
| Water (2) | 0.65 | 3.60 | 0.74 | 0.11 | 2.13 | 0.52 |
| Mean | 0.59 | 3.10 | 0.76 | 0.24 | 2.23 | 0.63 |

$R^2$: coefficient of determination, RMSE: Root Mean Square Error, and IoA: Index of Agreement [a] The number in parentheses in right of land use represents the numbers of weather stations for each land use.

Figure 8 shows the (a) observed $T_{MN}$ distribution and (b) predicted $T_{MN}$ distribution for a total of three days, Jan. 27, Jan. 30, and Feb. 7 in 2018, which were the days when the temperature sharply changed during the cold wave period. The temperature range was to −12 °C or less, which was the reference temperature of the cold wave, to show the distribution of cold wave occurrence. For all three days, the predicted $T_{MN}$ showed a similar distribution to that of the observed $T_{MN}$, but it was found that the range of cold wave occurrence was smaller than the observed $T_{MN}$. The trained data for the LSTM model were from 2008 to 2017, and the average minimum temperature increased by 0.24 °C during the period. Since the $T_{MN}$ predicted through the LSTM model reflected this upward trend, it seems that there was a difference in the distribution of $T_{MN}$ [33]. Figure 9 shows (a) the observed $T_{MX}$ distribution and (b) the predicted $T_{MX}$ distribution for three days, Jul. 20, Aug. 1, and Aug. 13, 2018. The temperature range was limited to 33 °C or more, which was the reference temperature of the heat wave, to show the distribution of heat wave occurrence. Overall, it was found that during the heat wave period, the distribution of heat wave occurrence was more similarly detected than during the cold wave period. However, even when the heat wave was detected, the temperature tended to be lower than the observed $T_{MX}$. This finding was partly an unavoidable result of model specifications, as the compared observations may be typical or extremely hot, while the $T_{MX}$ predictions were made for typical hot summer days [15].

In 2019, it was 0.5 °C hotter than normal years, and the heat wave period was 41% that in 2018 (31.4 days), lasting up to 13.3 days from late July to mid-August [49]. Additional verification of the SAT prediction model was performed on the duration of the heat wave (July 30 to August 11) where the longest heat wave occurred (Figure 10). On the other hand, in the winter of 2019, the national average minimum temperature was −1.4 °C, which was recorded as the warmest winter, and the cold wave period was just 0.4 days which is the fewest number cold wave days ever recorded since 1973 [50]. Due to the lack of data to verify the cold wave period, additional $T_{MN}$ verification was not conducted.

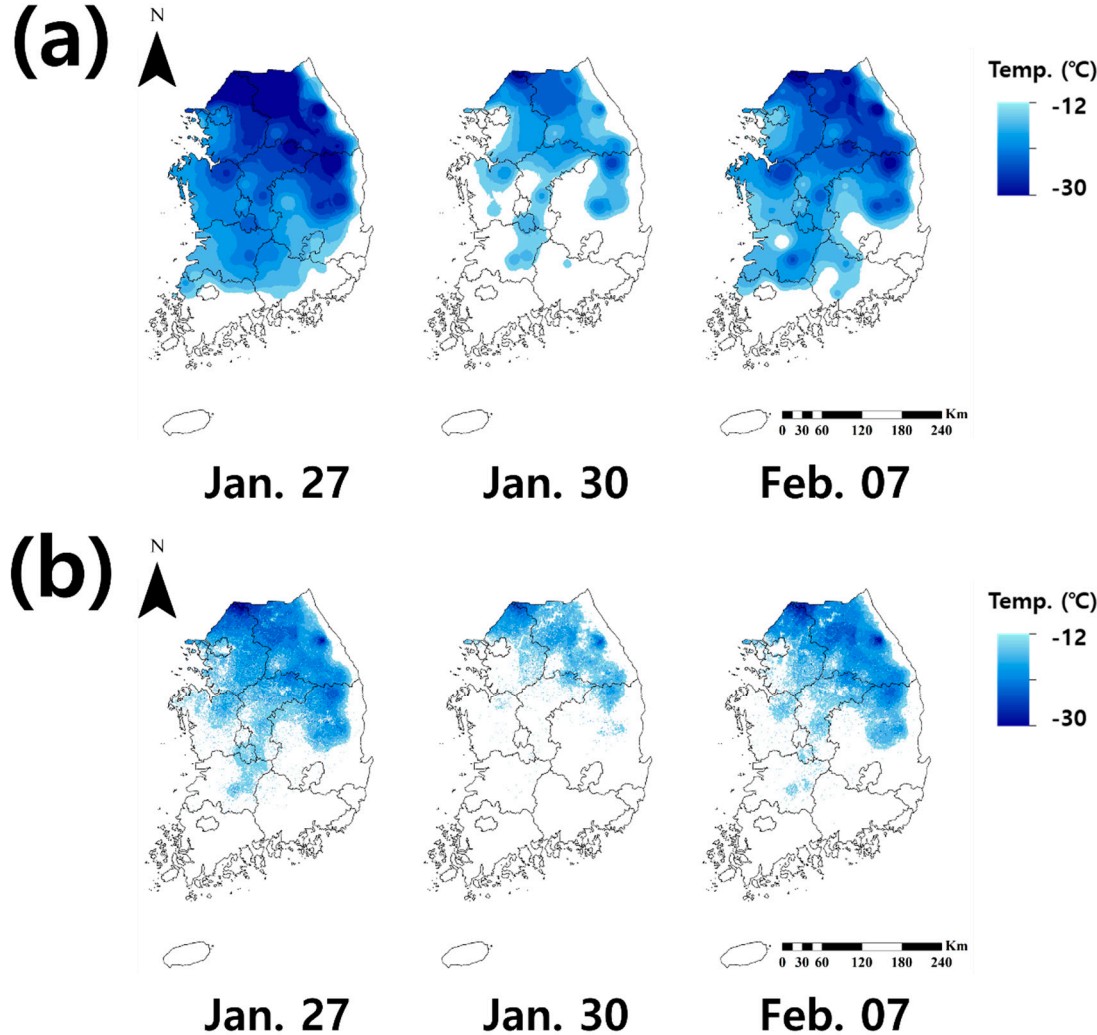

**Figure 8.** Spatial $T_{MN}$ distribution of (**a**) observed $T_{MN}$ and (**b**) predicted $T_{MN}$ on January 27, January 30, and February 7, 2018, during the cold wave period. The temperature range was set from −12 to −30 °C to express the distribution of cold wave occurrence.

Figure 10 shows the comparison of the mean observed $T_{MX}$ and mean predicted $T_{MX}$ from July 30 to August 11 for each land use scenario. Additionally, statistical indexes ($R^2$ and RMSE) were calculated and compared with the result of $T_{MX}$ prediction on the heat wave period in 2018. As shown in Figure 10, the RMSE tended to decrease compared to the SAT prediction for the 2018 heat wave period in all land uses except for the bare field. In the case of $R^2$, forest, grass, and bare fields decreased, and all other land uses are increased. The decrease in statistical indexes in bare field areas seems to be the main reason for the different temperature variations between the two stations. As mentioned earlier, since heatwaves are greatly affected by human activities such as land use change and increases in greenhouse gases, the predictive tendency in urban, rice paddy and upland crop areas seems to be relatively higher than in forest and grass areas. It is noteworthy that the wetlands show an excessively high correlation compared to other land uses. This seems to be the result of overfitting caused by training only with data from just one ASOS station located in the wetland.

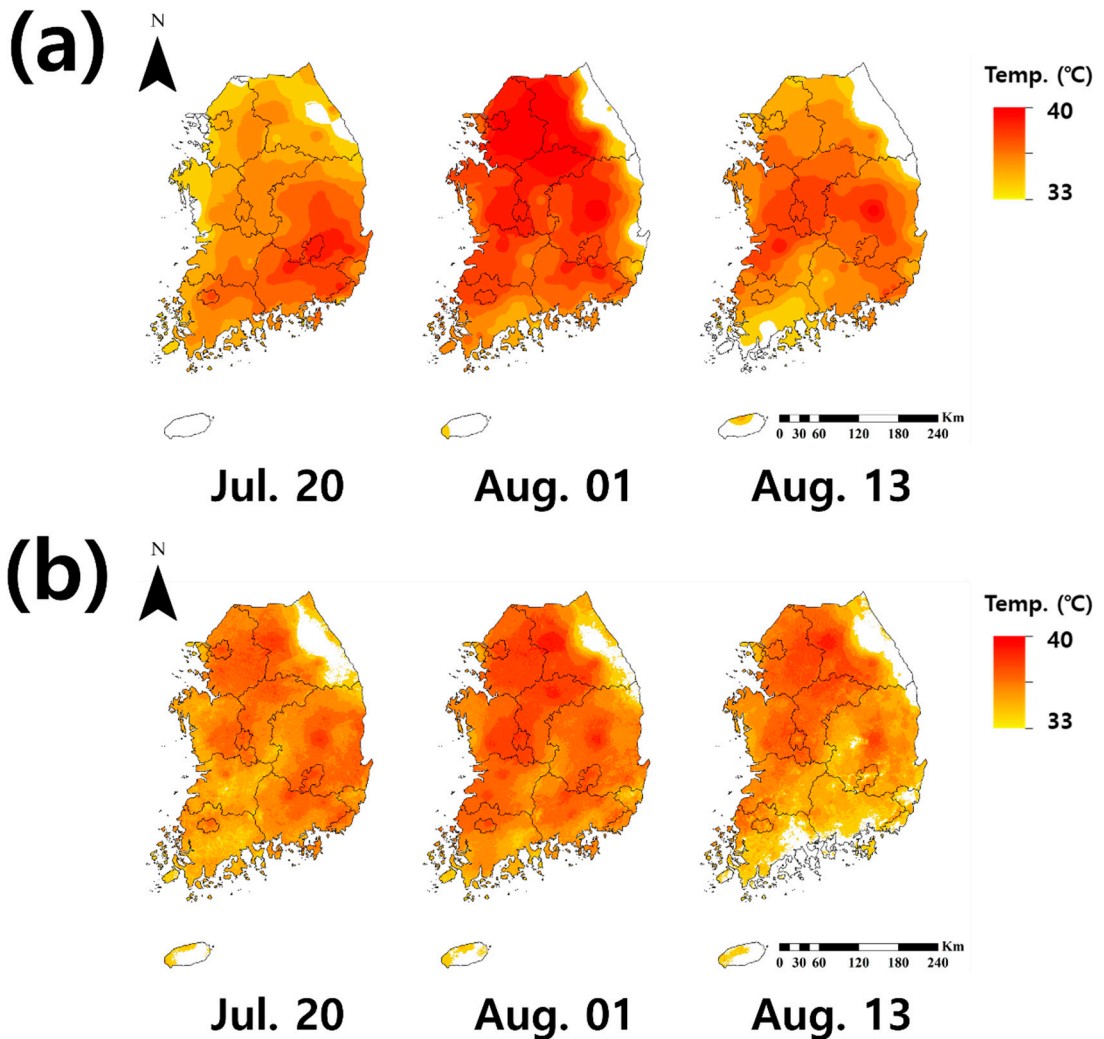

**Figure 9.** Spatial $T_{MX}$ distribution of (**a**) observed $T_{MX}$ and (**b**) predicted $T_{MX}$ on July 20, August 1, and August 13, 2018, during the heat wave period. The temperature range was set from 33 to 40 °C to express the distribution of heat wave occurrence.

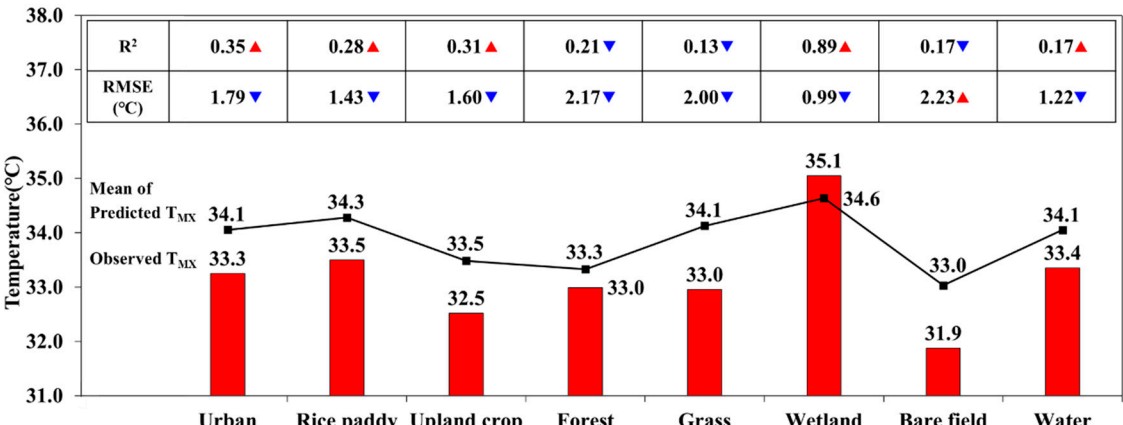

**Figure 10.** Comparison of mean observed $T_{MX}$ (red bar) and mean predicted $T_{MX}$ (solid line) on July 30 to August 11 for each land use classes. The triangle next to the upper statistical indicators ($R^2$: coefficient of determination; RMSE: Root Mean Square Error) shows the increase (red) and decrease (blue) compared to the result of $T_{MX}$ prediction on heat wave period in 2018.

## 4. Discussion

### 4.1. Correlation Analysis Results between MODIS LST and SAT and Usability for Regression Analysis

As a result of the correlation analysis between MODIS LST and SAT from 2008 to 2018, it was confirmed that the average value of daytime LST was closest to that of $T_{MX}$ and the average of nighttime LST was closet to $T_{MN}$. The daytime data of MODIS LST showed a larger variation in value compared to the nighttime data, which seems to be because the daytime data are more likely to be affected by changes in weather conditions due to the influx of solar radiation than the nighttime data (Figure 4). While with zero incoming solar radiance at night, the surface cools, and no noticeable temperature variation occurs among the stations because the heat output of the surface exceeds the input. Therefore, a sufficiently cooled surface (i.e., $LST_{AN}$) shows little spatial variation among the stations at night [16].

Overall, all 79 ASOS stations showed a similar tendency to appear earlier at the Chungju weather station in Figure 5 (Table 2). The *R* showed that nighttime LSTs were higher than the daytime LSTs, and especially for $T_{MM}$, the *R* was the highest with the nighttime LSTs. This result is because the nighttime LSTs show stable characteristics compared with the daytime LSTs. In general, the $T_{MX}$ of the day appears in the afternoon, and it might be expected that the daytime LST of Aqua MODIS ($LST_{AD}$) taken at 13:30 has the smallest deviation from the $T_{MX}$. However, there was no significant difference from the daytime LSTs at 4.8 °C. On the other hand, the difference from $T_{MM}$ and $T_{MN}$ showed that the Aqua MODIS LSTs were larger than the Terra MODIS LSTs, and this characteristic is well represented in Table 2. In a previous study [51], the seasonal RMSE analysis of AWS SAT and MODIS LST showed that the RMSE in winter was the lowest. In winter, the cold northwest monsoon blows during the Siberian anticyclone, which is the strongest semi-permanent high in the Northern Hemisphere, resulting in low temperature and humidity. Given that water vapor in the atmosphere plays an important role in the heat exchange process between the surface and atmosphere, low humidity in winter hinders heat and surface heat transfer and reduces the extent of winter SAT increases. In summary, the difference between the SAT and LST decreased. Moreover, due to the monsoon climatic characteristics of South Korea, the average cloud volume exceeds 50% in summer. Clouds are often observed in excess of 90%, especially from June to August [32]. As a result, the daily MODIS LST data often contained no valid cells or only a few cells. To solve this problem, gap-filling of the LST was performed, and the correlation with summer $T_{MX}$ decreased. According to the analysis, $LST_{AD}$ was the best data for the regression analysis during the heat wave period. However, since the deviation between each LST dataset was not large and the correlation was low, the use of MODIS LST for the regression analysis will require researchers to pay attention.

### 4.2. Climate Mitigation Effects during Heat and Cold Wave Periods

The results of Figure 7 might represent the climate mitigation effects of forests and wetlands. In the cold wave period, the average temperature of $LST_{TN}$ was approximately 2 to 5 °C higher than that of $T_{MN}$ because of the difference in measurement characteristics and time of each dataset. In addition, $T_{MN}$ generally showed higher average temperatures in paddy fields and wetlands, whereas LST showed a change in temperature characteristics by land use depending on the date (Appendix C).

In the heat wave period, the average wetland area was lower than that of the other land uses. Especially, in the $LST_{AD}$, the variation in the mean and maximum values for each land use increased, which made it possible to identify the climate mitigation effect of wetlands more clearly. In case of the other days of the heat wave period (see Appendix C), when the average and maximum values of urban areas were 40.0 °C and 51.5 °C, the wetlands were 36.5 °C and 44.6 °C, respectively, with 3.5 °C and 6.9 °C differences. The average and maximum values of paddy, upland crop, forest, grass, and water were lower than those of urban and bare fields, as shown in Figure 7b. Therefore, although the effect was different depending on the data analyzed, we could confirm the climate mitigation effect of the wetland and vegetation areas during the cold and heat wave periods.

Additioanlly, looking at the number of stations used in the analysis (Table 3), the number of stations in wetland, bare field, and water classes is one, two, and two, respectively, which are not enough sample numbers to indicate the characteristics of each land use. Therefore, in future studies, by adding AWS as well as ASOS, enough samples can be obtained to represent each land use to reliable research.

*4.3. Limitation and Improvement of Predicting SAT using LSTM with Remotely Sensed Variables*

Heat and cold waves can be caused by various factors (atmospheric pressure distribution, seasonal winds, foehn phenomenon, etc.) and it is hard to explain these phenomena solely with the relationship between SAT and LST. However, it cannot be denied that there is a certain relationship between SAT and LST. The SAT prediction using existing numerical weather prediction models such as the global data assimilation and prediction system (GDAPS), regional data assimilation and prediction system (RDAPS) of KMA (https://www.kma.go.kr), or other prediction models can produce more detailed and precise SAT data and inform its distribution, but the wider the spatio-temporal range to be predicted, the greater the amount of input data and computational resources are required. Therefore, this study aims to simplify the prediction of SAT—which can be complex—and evaluate its performance. The SAT predicted in this study had a similar spatial distribution with the observed SAT, but statistically, especially during the heatwave period, the average $R^2$ of all land use classes was 0.24, which was not good. Therefore, it is necessary to improve the performance of the SAT prediction model through an appropriate method.

As the climate mitigation effect on vegetation and water bodies was revealed in the SAT and LST characteristics analysis according to land use in Section 4.2, the accuracy of the model could be improved by adding vegetation and water indices such as NDVI, leaf area index (LAI), normalized difference water index (NDWI), and water ratio index (WRI) or topographical data (i.e. longitude, latitude, and elevation) as predictor variables to the SAT prediction. The addition of these related variables is also important for improving the performance of the model, but the indiscriminate addition of variables may lead to performance degradation. Therefore, among the variables, the robust variable should be selected to improve modeling SAT prediction at each point of the weather station and unknown locations [52].

Before adding some relative variables, to find out the robust variables among the four LST variables (LST$_{TD}$, LST$_{TN}$, LST$_{AD}$, and LST$_{AN}$) used in SAT prediction using TensorFlow-LSTM, the variable importance was evaluated (Table 4). To evaluate the variable importance, the normally sampled time-series data of each variable are taken to compute the model's prediction Y$_{norm}$. Then, randomly permuted data for each variable are utilized to calculate the prediction Y$_{rand}$ and the effect of perturbation, measured by calculating RMSE between Y$_{norm}$ and Y$_{rand}$ to evaluate the variable importance [52]. In variable importance measurement, the larger the difference in RMSE, the more important the variable is.

As shown in Table 4, although the most important variables were different for each land use, on average, it was found that LST$_{TD}$ and LST$_{AN}$ were important variables in the cold and heatwave period both as a value of 0.0500, 0.0642 for the cold wave period and 0.0664, 0.0738 for the heat wave period, respectively. In Section 3.1, it was found that LST$_{AN}$ had the best correlation during cold waves and LST$_{AD}$ during the heat waves. However, there was a slightly different pattern in the variable importance, because in Section 3.1, the correlation analysis was performed for all weather stations without considering land use, whereas the calculation of the importance of variables was performed considering land use. In addition, in Section 3.1, for the statistical results on the relationship between the SAT and LST on the same day, but with variable importance, the relationship between the predicted SAT one day later and the LST on the day is used, hence leading to a different result.

**Table 4.** Variable importance (root mean squared error difference between TensorFlow-LSTM model results using normal and randomly permuted samples) of the 4 LST variables used to predict SAT during the cold wave period and heat wave period considering land use characteristics.

| Land Use | Cold Wave Period (2018.01.23.~2018.02.13.) | | | | Heat Wave Period (2018.07.11.~2018.08.16.) | | | |
|---|---|---|---|---|---|---|---|---|
| | $LST_{TD}$ | $LST_{TN}$ | $LST_{AD}$ | $LST_{AN}$ | $LST_{TD}$ | $LST_{TN}$ | $LST_{AD}$ | $LST_{AN}$ |
| Urban | 0.0438 | 0.0354 | 0.0455 | 0.0443 | 0.0603 | 0.0765 | 0.0630 | 0.0784 |
| Rice paddy | 0.0695 | 0.0388 | 0.0667 | 0.0496 | 0.0630 | 0.0371 | 0.0395 | 0.0581 |
| Upland crop | 0.0616 | 0.0454 | 0.0295 | 0.0616 | 0.0767 | 0.0580 | 0.0466 | 0.0638 |
| Forest | 0.0568 | 0.0398 | 0.0444 | 0.0514 | 0.0776 | 0.0452 | 0.0397 | 0.0663 |
| Grass | 0.0445 | 0.0411 | 0.0363 | 0.0611 | 0.0708 | 0.0700 | 0.0389 | 0.0791 |
| Wetland | 0.0141 | 0.0469 | 0.0419 | 0.0821 | 0.0389 | 0.0803 | 0.0765 | 0.1200 |
| Bare field | 0.0405 | 0.0270 | 0.0282 | 0.0980 | 0.0716 | 0.0165 | 0.0393 | 0.0681 |
| Water | 0.0689 | 0.0495 | 0.0309 | 0.0656 | 0.0723 | 0.0393 | 0.0299 | 0.0563 |
| Mean | 0.0500 | 0.0405 | 0.0404 | 0.0642 | 0.0664 | 0.0529 | 0.0467 | 0.0738 |

$LST_{TD}$: Terra MODIS LST daytime, $LST_{TN}$: Terra MODIS LST nighttime, $LST_{AD}$: Aqua MODIS LST daytime, $LST_{AN}$: Aqua MODIS LST nighttime.

The implication in this Section is that as the mechanisms of SAT and LST are different for each land use scenario, variables to be used for prediction should be selected differently based on variable importance such as using wrapper feature selection methods, which are known as an intuitive and effective solution, that diminishes the number of subservient variables to extract subset showing the best performance in the modeling [53–56]. In a further study, it is expected that modeling SAT prediction, based on LST applying selective variables for each type of land use can improve prediction performance.

## 5. Conclusions

In this study, the characteristics of LST and SAT based on land use during heat and cold wave periods were analyzed, and spatial SAT was predicted using TensorFlow-LSTM with Terra and Aqua MODIS daytime and nighttime LSTs. The following points are summarized:

1.  As a result of the correlation analysis between SAT and LST, $LST_{TD}$ was well correlated with $T_{MX}$ (*R* 0.92 and RMSE 4.8 °C), and $LST_{TN}$ showed a good correlation with $T_{MN}$ (*R* 0.93 and RMSE 4.2 °C) from 2008 to 2018. For the analytical results of the cold and heat wave periods in 2018, $LST_{TN}$ showed suitable results for analysis with $T_{MN}$, where *R* was 0.60 and RMSE was 4.7 °C in the cold wave period, and $LST_{AD}$ was most correlated with $T_{MX}$, where *R* was 0.37 and RMSE was 5.4 °C during the heat wave period.

2.  Concerning the characteristics analysis of eight land use classes (urban, paddy, upland crop, forest, grass, wetland, bare field, and water) during the heat and cold wave periods, the climate mitigation effects of wetland and vegetation areas were confirmed. In the cold wave period, the average temperatures of urban and wetland areas were higher than those of other land covers because heat islands affect climate mitigating effects. During the heat wave period, the $T_{MX}$ was always reasonably above the heat wave reference temperature, while the $LST_{AD}$ was above or below the reference temperature. In addition, $T_{MX}$ did not show a significant difference in average temperature by land use, whereas $LST_{AD}$ showed a significant difference. Nevertheless, we could confirm the climate mitigation effect of wetlands and vegetation areas during heat and cold wave periods, although the effect was different depending on the data analyzed.

3.  The SAT prediction model using TensorFlow-LSTM was constructed for each of the eight land use classes for cold and heat wave periods. Each model simulated the $T_{MN}$ during the cold wave period and $T_{MX}$ during the heat wave period. As a result, during cold waves, the $T_{MN}$ prediction model had good explanatory power, with average values of $R^2$ of 0.59, RMSE of 3.10°C and IoA

of 0.76. In the comparison between the observed $T_{MN}$ and predicted $T_{MN}$ distribution, the model seems to reflect the trend of the annual average $T_{MN}$ rise due to climate change, and it was found that the model predicted the $T_{MN}$ as higher than the observed $T_{MN}$. During the heat wave period, the $T_{MX}$ prediction model was poorly described in comparison with the $T_{MN}$ prediction model, showing average values of $R^2$ and IoA of 0.24 and 0.63, respectively. However, RMSE was lower (2.23°C) than that of the $T_{MN}$ prediction model, and the change in the average annual $T_{MX}$ increase by climate change narrowed the difference between the observed $T_{MX}$ and predicted $T_{MX}$. The distribution of the predicted $T_{MX}$ compared with the observed $T_{MX}$ was distributed similarly to that of the cold wave period. However, because the observed $T_{MX}$ during heat waves are sometimes typical and sometimes extreme, the predicted $T_{MX}$ distribution tended to be lower as an unavoidable result. In addition, in the $T_{MX}$ and $T_{MN}$ prediction models, it was found that the existence of vegetation and water bodies for each land use influenced the prediction accuracy.

This study determined the possibility of remote sensing-based SAT prediction using MODIS LST and LSTM by land use classes. Using the results of this study, it will be possible to overcome the uncertainty of point observations by estimating the spatial distribution of SAT and predicting the occurrence area of heat and cold waves based on satellite data. In addition, the results of correlation analysis using long-term SAT and satellite data can be used as basic data in hydrological research. Although the accuracy of the spatial SAT distribution was somewhat low, a spatial distribution that shows a higher correlation can be created considering vegetation indexes, such as NDVI, EVI, and LAI, and topographical data, such as longitude, latitude, and elevation. This result is because the climate mitigation effects on vegetation and water areas appeared in the analyses of SAT and LST characteristics by land use. In addition, when the variable importance was calculated, the robust variables were different for each land use. In further studies, SAT predictions including these factors with feature selection will be conducted to improve the performance.

**Author Contributions:** Conceptualization, Y.L. and S.K.; Data Curation, J.C.; Software, W.J. and S.L.; Formal Analysis, J.C., Writing—Original Draft Preparation, J.C. and Y.L. and Writing—Review and Editing, Y.L. and S.K. All authors have read and agreed to the published version of the manuscript.

**Funding:** This subject is supported by the Korea Ministry of Environment (MOE) as "Water Management Research Program (79617)".

**Acknowledgments:** This manuscript was edited for English language by American Journal Experts (AJE).

**Conflicts of Interest:** The authors declare that they have no conflict of interest.

## Appendix A. Climate in South Korea

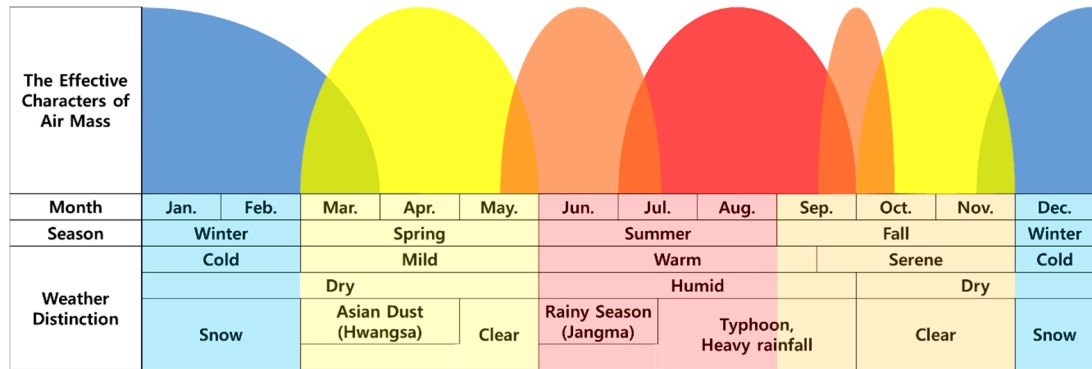

**Figure A1.** Seasonal climate characteristics of South Korea (https://www.kma.go.kr).

## Appendix B. Basic LSTM Equations

In this study, we used the LSTM equations as defined in [23]. The first stage of LSTM is to decide whether to forget or remember information at the cell state. The forget gate sigmoid layer makes this decision, and the output can be expressed as 0 (Forgotten) or 1 (Kept). The calculation formula is described as follows:

$$f_t = \sigma\left(W_f \cdot [h_{t-1}, x_t] + b_f\right) \tag{A1}$$

The second stage is to select information to be stored in the cell state. This stage is composed of two parts. First, the input gate sigmoid layer determines the value used for the update, and a *tanh* layer creates a vector of new candidate values $\widetilde{C}_t$, which can be added to the cell state. Second, these two layers are united to generate an update to the state. The calculation formula is described as follows:

$$i_t = \sigma(W_i \cdot [h_{t-1}, x_t] + b_i) \tag{A2}$$

$$\widetilde{C}_t = tanh(W_C \cdot [h_{t-1}, x_t] + b_C) \tag{A3}$$

In the third stage, the old cell state $C_{t-1}$ is updated by $f_t$ and $i_t * \widetilde{C}_t$. $C_{t-1}$ is multiplied by $f_t$ to remove unnecessary information; then, $i_t * \widetilde{C}_t$ is added to determine the new candidate value. This process will be scaled by how much we decide to update each state value. The calculation formula is described as follows:

$$C_t = f_t * C_{t-1} + i_t * \widetilde{C}_t \tag{A4}$$

The final stage is determining the LSTM's output. Through the sigmoid layer, the partial values of the cell state of the output are determined. The cell state is multiplied by the *tanh* function of the output of the sigmoid gate. The calculation formula is described as follows:

$$o_t = \sigma(W_o \cdot [h_{t-1}, x_t] + b_o) \tag{A5}$$

$$h_t = o_t * tanh(C_t) \tag{A6}$$

In Equations (A1)–(A6), $x_t$ is the input, and $h_{t-1}$ and $h_t$ are the outputs of the hidden layer. $\widetilde{C}_t$ is the input, and $C_t$ and $C_{t-1}$ are the outputs in the cell states. $f_t$, $i_t$ and $o_t$ are the outputs of the forget, input and output gates, respectively. The subscripts $t$ and in each input and output are time indicators. $W_f$, $W_i$, $W_o$ and $W_C$ are the weights linking $h_{t-1}$ and $x_t$ to each of the forget, input, and output gates and the cell state input, respectively. $b_f$, $b_i$, $b_o$ and $b_C$ are bias terms corresponding to each gate. $\sigma$ and *tanh* denote the functions of sigmoid $\frac{1}{1+\exp(-x)}$ and hyperbolic tangent $\frac{\exp(x)-\exp(-x)}{\exp(x)+\exp(-x)}$, respectively.

## Appendix C. Characteristics of SAT and LST at the Different Land Use

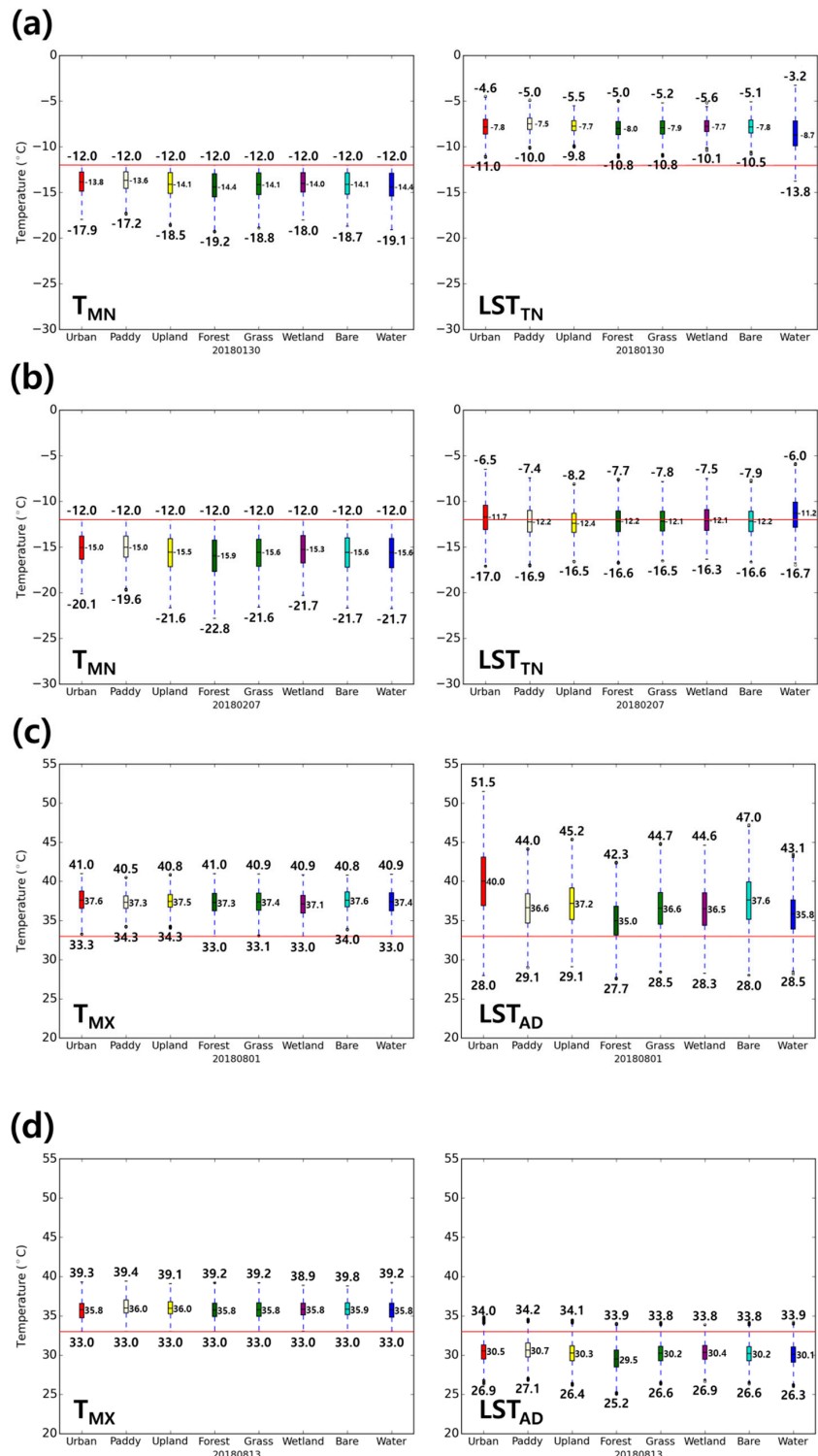

**Figure A2.** Boxplots of (**a**,**b**) observed $T_{MN}$ and $LST_{TN}$ for 8 land use classes (red: urban, light yellow: paddy, yellow: upland crop, dark green: forest, green: grass, purple: wetland, cyan: bare field, and dark blue: water) on January 30 and February 7, 2018, respectively, and (**c**,**d**) observed $T_{MX}$ and $LST_{AD}$ on August 1 and 13, 2018, respectively. In (**a**,**b**), the red solid line on the graph is −12 °C, which represents the reference temperature of the cold wave and in (**c**,**d**), the red solid line on the graph is 33 °C, which represents the reference temperature of the heat wave.

## Appendix D. SAT Prediction Model Result

**Table A1.** The $T_{MN}$ prediction model results from 79 meteorological stations during the cold wave period (2018.01.23.–2018.02.13.). The unit of RMSE is degrees Celsius.

| STN | LU | $R^2$ | RMSE | IoA | STN | LU | $R^2$ | RMSE | IoA | STN | LU | $R^2$ | RMSE | IoA |
|---|---|---|---|---|---|---|---|---|---|---|---|---|---|---|
| 90 | UL | 0.71 | 2.55 | 0.68 | 159 | UB | 0.57 | 5.26 | 0.58 | 273 | UL | 0.62 | 2.88 | 0.64 |
| 95 | UB | 0.74 | 4.28 | 0.61 | 165 | WT | 0.66 | 2.97 | 0.44 | 277 | FR | 0.50 | 2.73 | 0.72 |
| 98 | UB | 0.74 | 2.15 | 0.75 | 172 | PD | 0.64 | 2.19 | 0.52 | 278 | PD | 0.83 | 4.35 | 0.65 |
| 100 | UL | 0.52 | 3.17 | 0.57 | 174 | UL | 0.63 | 2.93 | 0.54 | 279 | UB | 0.44 | 4.14 | 0.74 |
| 101 | UB | 0.32 | 6.55 | 0.70 | 175 | FR | 0.44 | 2.32 | 0.48 | 281 | PD | 0.60 | 2.76 | 0.66 |
| 104 | FR | 0.51 | 2.58 | 0.73 | 192 | UB | 0.58 | 4.43 | 0.63 | 284 | PD | 0.61 | 2.98 | 0.56 |
| 105 | UB | 0.55 | 3.51 | 0.74 | 202 | UB | 0.37 | 5.26 | 0.66 | 285 | FR | 0.56 | 3.18 | 0.67 |
| 106 | BR | 0.46 | 2.43 | 0.74 | 203 | FR | 0.69 | 2.43 | 0.63 | 288 | UB | 0.43 | 5.30 | 0.76 |
| 108 | UB | 0.52 | 3.13 | 0.64 | 211 | GR | 0.70 | 4.91 | 0.70 | 289 | UL | 0.51 | 2.98 | 0.61 |
| 112 | UB | 0.43 | 3.63 | 0.61 | 212 | UB | 0.69 | 2.70 | 0.65 | 294 | UB | 0.57 | 4.40 | 0.65 |
| 114 | UB | 0.64 | 3.64 | 0.66 | 216 | UB | 0.44 | 4.52 | 0.65 | 295 | FR | 0.67 | 2.23 | 0.70 |
| 119 | UB | 0.56 | 3.45 | 0.63 | 221 | UL | 0.73 | 3.91 | 0.69 | 217 | FR | 0.56 | 2.69 | 0.64 |
| 121 | GR | 0.70 | 3.89 | 0.69 | 226 | UL | 0.77 | 3.7 | 0.62 | 252 | UL | 0.59 | 2.31 | 0.57 |
| 127 | GR | 0.65 | 4.53 | 0.64 | 232 | UL | 0.73 | 2.52 | 0.63 | 253 | UB | 0.58 | 4.97 | 0.61 |
| 129 | FR | 0.64 | 2.03 | 0.63 | 235 | PD | 0.55 | 2.59 | 0.56 | 254 | UB | 0.51 | 4.14 | 0.65 |
| 131 | UB | 0.67 | 3.24 | 0.66 | 236 | PD | 0.56 | 4.05 | 0.64 | 255 | UB | 0.59 | 5.18 | 0.69 |
| 133 | WT | 0.64 | 4.23 | 0.61 | 238 | UL | 0.78 | 3.27 | 0.65 | 257 | BR | 0.62 | 2.08 | 0.67 |
| 135 | UL | 0.61 | 3.23 | 0.62 | 243 | GR | 0.64 | 3.92 | 0.57 | 258 | PD | 0.49 | 3.08 | 0.68 |
| 136 | FR | 0.68 | 2.38 | 0.67 | 244 | UB | 0.73 | 2.49 | 0.59 | 259 | UB | 0.49 | 3.89 | 0.46 |
| 137 | UB | 0.61 | 3.51 | 0.61 | 245 | FR | 0.69 | 2.33 | 0.59 | 263 | UB | 0.45 | 5.12 | 0.75 |
| 138 | UB | 0.54 | 4.75 | 0.76 | 247 | FR | 0.57 | 4.36 | 0.61 | 264 | UB | 0.47 | 2.50 | 0.65 |
| 140 | UB | 0.56 | 4.41 | 0.60 | 248 | UL | 0.70 | 3.50 | 0.51 | 266 | UB | 0.51 | 5.31 | 0.76 |
| 143 | UB | 0.38 | 4.09 | 0.79 | 260 | GR | 0.43 | 3.79 | 0.47 | 268 | GR | 0.55 | 2.49 | 0.41 |
| 146 | UB | 0.49 | 3.93 | 0.61 | 261 | PD | 0.56 | 3.33 | 0.50 | 276 | UB | 0.47 | 3.82 | 0.68 |
| 152 | UB | 0.59 | 4.53 | 0.76 | 262 | UB | 0.29 | 4.45 | 0.47 | 283 | WL | 0.50 | 2.09 | 0.69 |
| 155 | FR | 0.60 | 2.51 | 0.60 | 271 | PD | 0.66 | 3.30 | 0.65 | **Mean** | | **0.58** | **3.52** | **0.63** |
| 156 | UB | 0.63 | 3.52 | 0.67 | 272 | GR | 0.51 | 3.38 | 0.63 | | | | | |

STN: station number, LU: sand use class (UB: urban, PD: paddy area, UL: upland crop, FR: forest, GR: grass, WL: wetland, BR: bare, WT: Water), $R^2$: coefficient of determination, RMSE: root mean square error (°C), IoA: index of agreement.

**Table A2.** The $T_{MX}$ prediction model results of 79 meteorological stations in the heat wave period (2018.07.11.–2018.08.16.). The unit of RMSE is degrees Celsius.

| STN | LU | $R^2$ | RMSE | IoA | STN | LU | $R^2$ | RMSE | IoA | STN | LU | $R^2$ | RMSE | IoA |
|---|---|---|---|---|---|---|---|---|---|---|---|---|---|---|
| 90 | UL | 0.35 | 3.85 | 0.83 | 159 | UB | 0.15 | 2.21 | 0.60 | 273 | UL | 0.17 | 2.04 | 0.82 |
| 95 | UB | 0.43 | 2.82 | 0.77 | 165 | WT | 0.05 | 2.33 | 0.78 | 277 | FR | 0.48 | 2.99 | 0.78 |
| 98 | UB | 0.44 | 2.01 | 0.84 | 172 | PD | 0.11 | 1.96 | 0.87 | 278 | PD | 0.24 | 1.88 | 0.71 |
| 100 | UL | 0.15 | 3.29 | 0.81 | 174 | UL | 0.09 | 1.67 | 0.74 | 279 | UB | 0.33 | 1.76 | 0.59 |
| 101 | UB | 0.42 | 2.31 | 0.54 | 175 | FR | 0.25 | 3.01 | 0.81 | 281 | PD | 0.25 | 2.64 | 0.78 |
| 104 | FR | 0.39 | 3.09 | 0.80 | 192 | UB | 0.16 | 1.84 | 0.63 | 284 | PD | 0.12 | 1.77 | 0.74 |
| 105 | UB | 0.36 | 2.91 | 0.75 | 202 | UB | 0.31 | 2.34 | 0.60 | 285 | FR | 0.22 | 1.92 | 0.72 |
| 106 | BR | 0.37 | 2.26 | 0.79 | 203 | FR | 0.24 | 2.36 | 0.84 | 288 | UB | 0.34 | 1.69 | 0.55 |
| 108 | UB | 0.30 | 2.55 | 0.73 | 211 | GR | 0.42 | 2.82 | 0.65 | 289 | UL | 0.15 | 2.02 | 0.69 |
| 112 | UB | 0.23 | 2.25 | 0.69 | 212 | UB | 0.35 | 2.81 | 0.85 | 294 | UB | 0.21 | 1.77 | 0.61 |
| 114 | UB | 0.28 | 2.31 | 0.74 | 216 | UB | 0.26 | 2.89 | 0.66 | 295 | FR | 0.27 | 1.26 | 0.81 |
| 119 | UB | 0.21 | 2.30 | 0.70 | 221 | UL | 0.30 | 2.09 | 0.74 | 217 | FR | 0.26 | 3.10 | 0.84 |
| 121 | GR | 0.33 | 2.51 | 0.70 | 226 | UL | 0.17 | 1.71 | 0.76 | 252 | UL | 0.07 | 1.57 | 0.87 |
| 127 | GR | 0.25 | 2.37 | 0.63 | 232 | UL | 0.27 | 2.01 | 0.86 | 253 | UB | 0.13 | 2.19 | 0.66 |
| 129 | FR | 0.30 | 2.00 | 0.82 | 235 | PD | 0.22 | 2.04 | 0.73 | 254 | UB | 0.17 | 1.77 | 0.67 |
| 131 | UB | 0.20 | 1.68 | 0.76 | 236 | PD | 0.27 | 1.96 | 0.64 | 255 | UB | 0.28 | 1.80 | 0.60 |
| 133 | WT | 0.16 | 1.94 | 0.70 | 238 | UL | 0.23 | 1.61 | 0.77 | 257 | BR | 0.27 | 1.90 | 0.85 |

**Table A2.** *Cont.*

| STN | LU | $R^2$ | RMSE | IoA | STN | LU | $R^2$ | RMSE | IoA | STN | LU | $R^2$ | RMSE | IoA |
|-----|----|-------|------|-----|-----|----|-------|------|-----|-----|----|-------|------|-----|
| 135 | UL | 0.16 | 1.82 | 0.76 | 243 | GR | 0.11 | 1.93 | 0.70 | 258 | PD | 0.24 | 1.29 | 0.67 |
| 136 | FR | 0.27 | 1.94 | 0.85 | 244 | UB | 0.11 | 1.64 | 0.86 | 259 | UB | 0.20 | 2.57 | 0.63 |
| 137 | UB | 0.21 | 2.35 | 0.71 | 245 | FR | 0.10 | 1.78 | 0.84 | 263 | UB | 0.37 | 1.52 | 0.60 |
| 138 | UB | 0.38 | 3.02 | 0.65 | 247 | FR | 0.14 | 1.67 | 0.66 | 264 | UB | 0.21 | 1.54 | 0.80 |
| 140 | UB | 0.18 | 1.94 | 0.62 | 248 | UL | 0.04 | 1.80 | 0.75 | 266 | UB | 0.35 | 1.65 | 0.57 |
| 143 | UB | 0.42 | 2.09 | 0.62 | 260 | GR | 0.26 | 2.66 | 0.62 | 268 | GR | 0.03 | 1.90 | 0.70 |
| 146 | UB | 0.11 | 1.82 | 0.68 | 261 | PD | 0.18 | 1.92 | 0.67 | 276 | UB | 0.29 | 2.30 | 0.72 |
| 152 | UB | 0.35 | 1.95 | 0.66 | 262 | UB | 0.16 | 1.91 | 0.58 | 283 | WL | 0.35 | 2.79 | 0.82 |
| 155 | FR | 0.17 | 1.72 | 0.81 | 271 | PD | 0.29 | 2.42 | 0.83 | **Mean** | | 0.24 | 2.15 | 0.72 |
| 156 | UB | 0.18 | 1.71 | 0.71 | 272 | GR | 0.22 | 2.28 | 0.71 | | | | | |

STN: station number, LU: land use class (UB: urban, PD: paddy area, UL: upland crop, FR: forest, GR: grass, WL: wetland, BR: bare, WT: water), $R^2$: coefficient of determination, RMSE: root mean square error (°C), IoA: index of agreement.

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
