# Peer review of "Correlation Analysis between Air Temperature and MODIS Land Surface Temperature and Prediction of Air Temperature Using TensorFlow Long Short-Term Memory for the Period of Occurrence of Cold and Heat Waves"

_remotesensing, doi:10.3390/rs12193231_

Round 1
Reviewer 1 Report
The research is very interesting, but there are a few points still to be improved for the following points:
1) all equations should be cited for references if they are not developed by authors;
2) A section discussion should be added before conclusion.
More specifically:
1) Line 110, "all regions of South Korea (33°8'N to 39°0'N and 124°5'E to 130°0'E)" should be "all regions of South Korea (124°5'E to 130°0'E to 33°8'N to 39°0'N)";
2) validation should also be mentioned in results;
3) lastest references should be updated if applicable.
In short, it needs a revision as mentioned above.
Author Response
The research is very interesting, but there are a few points still to be improved for the following points:
: The manuscript has been modified to reflect the points as you pointed out. Thank you for your advice on making it a better paper.
1) all equations should be cited for references if they are not developed by authors;
: (Line 563-586, Eq. B1-B5) Added references for all equations and moved to Appendix B.
2) A section discussion should be added before conclusion.
: (Line 403-504) The results and discussion were separated, and the additional analysis are summarized in the discussion.
More specifically:
1) Line 110, "all regions of South Korea (33°8'N to 39°0'N and 124°5'E to 130°0'E)" should be "all regions of South Korea (124°5'E to 130°0'E to 33°8'N to 39°0'N)";
: (Line 103) The sentence was revised as you mentioned.
2) validation should also be mentioned in results;
: The verification of the predicted SAT is written in Table 3 and the Appendix D.
3) lastest references should be updated if applicable.
: (Reference number 17, 18, 39, and 51 to 55) The latest references have been added.

Reviewer 2 Report
General comments:
The manuscript compares MODIS LST with SAT from weather stations in South Korea. Primary focus is on a heat wave and on a cold wave and on using machine learning (LSTM) to model spatial SATs. The topic is interesting and relevant, and the article is generally well written. However, the topic is partially very repetitive to previous work (that can be fixed by simply focusing more on the new elements) and there are quite a bit of inconsistencies and structural problems, hence I do recommend some revisions and some cleaning up the manuscript before accepting this manuscript for publication.
Specific major comments:
- The manuscript is pretty messy particularly figures and tables. E.g. there are two table 1s!!!! The Figures are quite repetitive, just show me the relevant stuff but in a clearer way (see specific comments). There are also to many unnecessary figures e.g. Fig 3 and 7 might be more suited for a supplement or appendix.
- While the manuscript goes into a lot of details when describing the exact methods (e.g. a paragraph describing spearman coefficients and a ton of detailed information about LSTM), the overall picture is missing. Particularly when describing the TensorFlow-LSTM a short sentence just stating in layman's terms what this method does would be useful. I assume not all climatologist/other researchers reading this article will be familiar with machine learning.
- The most interesting part of this study is probably the modeling of spatial SAT, while this is also promoted in abstract and introduction, the results only talk about this very briefly compared to the (too long) discussion of comparing SAT and LST.
- You are comparing day and nighttime LST from Terra and Aqua to min and max SAT – why not to SAT of the time of overflight (10:30 and 1:30 am and pm)?
- MODIS has a 1 km resolution and SAT is simply measured at any point withing this pixel. Are there pixels with 2 SAT stations? Do you interpolate to get LAT at the point location?
- Also curios about the landcover. Apparently, you are using a SK one that has a 1 km resolution – are these pixels resampled to fit with the modis pixels? I’m primarily surprised by your analysis of LST over water – are these weather stations over water? LST is also typically not available for water surface.
Specific minor comments:
- Title – not a big fan of the abbreviations in the title (LST and LSTM)
- Introduction – the motivation you are describing in your first paragraph is a bit strange, I assume you mean to say that heat waves and cold waves a re bad but we do not have enough weather stations to properly monitor them in SAT, hence modeling them from LST is useful. This massage is currently a bit convoluted, e.g. by including 5 lines about past weather (line 55-61). I would recommend deleting these lines and in generally just making sure each paragraph has a clear massage. The last one (98 – 105) is really good and clear!
- Line 121 – how did you define poor-quality stations?
- Line 124. What year is the Land cover from. Are there landcover changes?
- Line 142 – you mean SAT? I was sometimes confused which temperatures you were talking. I’d make sure to never just use temperature but alwas SAT or LST.
- Table 1 – how many stations are there in each landcover class?
- Figure 2 – get rid of the empty box. Generally, your figures are a bit messy. Try cleaning up text size. Right now, some of the righting is ridiculously small. And every textbox seems a different size. This is relevant for all figures
- Second Table 1- just call it SAT (it is surface air temperature at 2m height, is it?) min, max and mean and LST 1:30 am and pm or 10:30am and pm. Currently its honestly a bit confusing. That way you also don’t need this table.
- Line 176 ff – I’m not super familiar with this sort of machine learning and needed to google quite a few basics to be able to understand what you were doing. I would recommend a very basic sentence at the beginning just explaining what LSTM does (Modeling SAT of t_i based on an LST timeseries for a location x?). Also just to make sure, you did not develop this method, did you? You give a lot of info here that could be used to supplements/appendix
- Chapter 3.1 is too long considering that comparing SAT and LST has been done quite often before. What is the key massage here? Aslo not sure why you chose to compare. I don’t think its necessary to compare Nighttime LST to SATmax.
- Fig 6 if feel there is too much info. Put most of it inot the supplement and simply show a seasonal (one average year) comparison of SAT and LST. What do you think about doing a diurnal comparison (you have 4 LSTs a day from MODIS
- Chapter 3.2. – I would be interested in an analysis by land use during normal times as well. Particularly in the offset between SAT and LST which is not really included in here at all. You claim this confirms the climate mitigation effects of evapotranspiration which of course exists, but I don’t really see it in your Figures 9 and 10. By combining data from the entire nation your boxplots cover such a temperature span making it hard to see any sort of (significant) difference. However, this is also not very new, so no need to discuss it in too many figures
- Chapter 3.3. this is by far the most interesting part of your work and at the same time the only one missing some detail. I would prefer also seeing a time series of your model. I assumed you did train on all weather stations. Did you try something like bootstrapping to get a better feel for the goodness of you model? Does topography play into how well the model works. I’m also nut sure if it is different models for the heat wave and cold wave or if it is one model simply applied to different times. What kind of features appeared to be relevant in the machine learning? Is cloud cover an issue?
- Fig 11 – SAT here are interpolated – why? I would suggest showing it only at the weather stations or also only at the weather stations. What is the benefit from your model to a simple interpolation? Also would like to see a map of LST.
Author Response
General comments:
The manuscript compares MODIS LST with SAT from weather stations in South Korea. Primary focus is on a heat wave and on a cold wave and on using machine learning (LSTM) to model spatial SATs. The topic is interesting and relevant, and the article is generally well written. However, the topic is partially very repetitive to previous work (that can be fixed by simply focusing more on the new elements) and there are quite a bit of inconsistencies and structural problems, hence I do recommend some revisions and some cleaning up the manuscript before accepting this manuscript for publication.
: The manuscript has been modified to reflect the points as you pointed out. Thank you for your advice on making it a better paper.
Specific major comments:
The manuscript is pretty messy particularly figures and tables. E.g. there are two table 1s!!!! The Figures are quite repetitive, just show me the relevant stuff but in a clearer way (see specific comments). There are also to many unnecessary figures e.g. Fig 3 and 7 might be more suited for a supplement or appendix.
: We corrected the numbers of all tables and figures appropriately and edited them to show only the necessary information. In addition, Figure 3 has been moved to the Appendix A, and Figure 7 has been deleted because the information is similar to that of Table 2.
While the manuscript goes into a lot of details when describing the exact methods (e.g. a paragraph describing spearman coefficients and a ton of detailed information about LSTM), the overall picture is missing. Particularly when describing the TensorFlow-LSTM a short sentence just stating in layman's terms what this method does would be useful. I assume not all climatologist/other researchers reading this article will be familiar with machine learning.
: (Line 171-173) At the beginning of Section 2.3, the contents performed in this study are described in an easy-to-understand manner as follows:
“In this study, predicting the SAT at a time t based on an LST time-series dataset at a time t-1 at a point location of each weather station was performed with LSTM, an advanced form of RNN, one of the deep learning algorithms.”
The most interesting part of this study is probably the modeling of spatial SAT, while this is also promoted in abstract and introduction, the results only talk about this very briefly compared to the (too long) discussion of comparing SAT and LST.
: (Line 455-504) We added discussion on SAT prediction.
You are comparing day and nighttime LST from Terra and Aqua to min and max SAT – why not to SAT of the time of overflight (10:30 and 1:30 am and pm)?
: One of the objectives of this study is to analyze the characteristics between the daily maximum and minimum SAT and LST. As you said, we can use the minute-by-minute data at the time of overflight, but the minute-by-minute data cannot be considered to reflect the characteristics of the highest/lowest value of the temperature on that day. Therefore, we used daily maximum and minimum SAT data in this study.
MODIS has a 1 km resolution and SAT is simply measured at any point withing this pixel. Are there pixels with 2 SAT stations? Do you interpolate to get LAT at the point location?
: As can be seen from Figure 2, there are no two weather stations located within 1 km. In addition, the observation density of ASOS, which is managed by the Korean Meteorological Administration (KMA), is 67 km, and since it is 13 km including AWS, the two stations do not overlap within 1 km.
The following is the introduction page of the ground weather observation network of the KMA: http://www.kma.go.kr/aboutkma/biz/observation02.jsp
Also curios about the landcover. Apparently, you are using a SK one that has a 1 km resolution – are these pixels resampled to fit with the modis pixels? I’m primarily surprised by your analysis of LST over water – are these weather stations over water? LST is also typically not available for water surface.
: In this study, the level-2 land use with a spatial resolution of 10m produced in 2013 was resampled and used with a resolution of 1000m. The maximum likelihood method is used when resamples land use data with a resolution of 30m to 1000m. If the water area dominates the land use around the station, the land use of the station is designated as the water area. The meteorological station is not really located on the water and LST is not typically applicable to the water surface, but we wanted to see what mechanism it exhibits according to the dominant land use near the weather station.
Specific minor comments:
Title – not a big fan of the abbreviations in the title (LST and LSTM)
: (Line 2-6) We have changed the title to “Correlation Analysis between Air Temperature and MODIS Land Surface Temperature and Prediction of Air Temperature Using TensorFlow Long Short-Term Memory for the Period of Occurrence of Cold and Heat Waves”.
Introduction – the motivation you are describing in your first paragraph is a bit strange, I assume you mean to say that heat waves and cold waves are bad but we do not have enough weather stations to properly monitor them in SAT, hence modeling them from LST is useful. This message is currently a bit convoluted, e.g. by including 5 lines about past weather (line 55-61). I would recommend deleting these lines and in generally just making sure each paragraph has a clear message. The last one (98 – 105) is really good and clear!
: As you suggested, we deleted line (55-61) to clarify and concise the meaning of the paragraph.
Line 121 – how did you define poor-quality stations?
: (Line 113-116) We defined observatories that do not have data corresponding to the application period (2008-2018) of this study and observatories that are out of the range of land use map as poor-quality stations. However, it was determined that the choice of the term poor-quality stations could confuse readers, so it was deleted and described as follows:
“Therefore, for the observed maximum, minimum, and mean SAT (TMX, TMN, and TMM) data used in this study, 79 stations of total ASOS station data were used daily from 2008 to 2018, except for the stations with no data for the period or out of the range of land use map and AWSs (Figure 2a).”
Line 124. What year is the Land cover from. Are there landcover changes?
: In this study, the level-2 land use map produced in 2013 was used. A new land use map was produced in 2019, but since this study conducted analysis and learning using past data, land use data from 2013 were used. Although it is possible to analyze changes in land use, it was not considered to focus on the purpose of the study. In the future, it is planned to analyze heat waves according to changes in land use.
Line 142 – you mean SAT? I was sometimes confused which temperatures you were talking. I’d make sure to never just use temperature but always SAT or LST.
: We have unified the terms SAT and LST by reviewing the entire manuscript.
Table 1 – how many stations are there in each landcover class?
: (Table 3) We added the number of stations for each land use. In Table 3, the number of stations in Wetland, Bare field, and Water classes is 1, 2, and 2, respectively, which are not enough sample numbers to indicate the characteristics of each land use. Therefore, in future studies, by adding AWS as well as ASOS, it can be enough samples to represent each land use to reliable research.
Figure 2 – get rid of the empty box. Generally, your figures are a bit messy. Try cleaning up text size. Right now, some of the righting is ridiculously small. And every textbox seems a different size. This is relevant for all figures
: We matched font sizes in all figures similarly and made it a little bit larger.
Second Table 1- just call it SAT (it is surface air temperature at 2m height, is it?) min, max and mean and LST 1:30 am and pm or 10:30am and pm. Currently its honestly a bit confusing. That way you also don’t need this table.
: (Line 114 and 159) We deleted the table and made the abbreviation explained in the manuscript.
Line 176 ff – I’m not super familiar with this sort of machine learning and needed to google quite a few basics to be able to understand what you were doing. I would recommend a very basic sentence at the beginning just explaining what LSTM does (Modeling SAT of t_i based on an LST timeseries for a location x?). Also just to make sure, you did not develop this method, did you? You give a lot of info here that could be used to supplements/appendix
: (Line 171-173 and 185-196) As you said, LSTM is not my development. Therefore, we moved it to the appendix after the citation of the basic formula. In addition, the following explanations related to LSTM have been described to make it easier for readers to understand.
“In this study, predicting the SAT at a time t based on an LST time-series dataset at a time t-1 at a point location of each weather station was performed with LSTM, an advanced form of RNN, one of the deep learning algorithms.”
“The advantage of a recurrent neural network is that it can reflect previous information to the current information. However, to solve the long-term dependence problem, it is necessary to remember not only previous short-term memory but also long-term memory. Therefore, in LSTM, information is stored and used in a separate Cell Ct. Also, since the information stored in Ct must be forgotten if it is not related to the current information, the Ct, and the information to be output ht are separately calculated. The gates used here to judge the information to remember or whether forget are the input gate and the forget gate. In LSTM, through this forget gate, it is possible to prevent the problem that the computational amount may explode using all the past information in the RNN by cutting unnecessary information.”
Chapter 3.1 is too long considering that comparing SAT and LST has been done quite often before. What is the key massage here? Also not sure why you chose to compare. I don’t think its necessary to compare Nighttime LST to SATmax.
: Section 3.1 shows that there is a relationship between the two by comparing the SAT and LST and finding out which LST is most correlated with each SAT during the entire period, heatwave, and cold wave period. In other words, the main message in this section is that since each SAT has a different LST, which is highly correlated, the data used for prediction and analysis must also be different. In general, the maximum SAT and nighttime LST have a large difference in temperature and different time zones, but the reason for comparing 2 temperature is because there were cases where the LST in the nighttime had more correlations because of the less fluctuation than the LST in the daytime. However, as you mentioned, we drew a graph comparing TMX and daytime LSTs, and TMN and nighttime LSTs for more relevant comparison. When drawing a graph for the entire period, the characteristics of each data were not clearly visible, so the graph was changed to a seasonal graph for the average year, and statistics such as R2 are shown in Table 2 and deleted.
Fig 6 if feel there is too much info. Put most of it into the supplement and simply show a seasonal (one average year) comparison of SAT and LST. What do you think about doing a diurnal comparison (you have 4 LSTs a day from MODIS)
: (Figure 5) A Time Series graph was created for 2013 to enable seasonal comparison. In addition, the composition of the graph was changed to compare highly correlated data, TMX and daytime LSTs, and TMN and nighttime LSTs, and the symbology of LST was changed from line to hollow dot.
Chapter 3.2. – I would be interested in an analysis by land use during normal times as well. Particularly in the offset between SAT and LST which is not really included in here at all. You claim this confirms the climate mitigation effects of evapotranspiration which of course exists, but I don’t really see it in your Figures 9 and 10. By combining data from the entire nation your boxplots cover such a temperature span making it hard to see any sort of (significant) difference. However, this is also not very new, so no need to discuss it in too many figures
: (Line 277-339 and 436-454) In the Boxplot of this paper (Figure 7), climate mitigation effects for each land use do not appear dramatically. However, rather than focusing on the climate mitigation effect, this is to show that the characteristics of each land use are different during the heatwave and cold wave period so that the classification by land use is necessary for SAT prediction. As you said, the contents and figures are summarized, and only necessary parts are described.
Chapter 3.3. this is by far the most interesting part of your work and at the same time the only one missing some detail. I would prefer also seeing a time series of your model. I assumed you did train on all weather stations. Did you try something like bootstrapping to get a better feel for the goodness of you model? Does topography play into how well the model works. I’m also not sure if it is different models for the heat wave and cold wave or if it is one model simply applied to different times. What kind of features appeared to be relevant in the machine learning? Is cloud cover an issue?
: (Line 455-504) In this study, since the spatial SAT for about one month was predicted for both cold and heat waves after training, I did not draw it because I thought that the size of the picture would be smaller and the distribution would be difficult to see if expressed as a time series.
The training was conducted based on data from all stations and Bootstrapping was not considered. The influence of the topography was not considered, but we described in Section 4.3 because it is thought that it can be used in future studies to improve the results.
The model used is a model trained on the highest and lowest temperatures for each land use based on past observations (2008~2017) and applied to the period of the heatwave and cold waves.
The most relevant factors for machine learning were different for each of the highest and lowest temperatures and land use when variable importance was calculated, and the results are described in Section 4.3. Overall, it was found that the importance of LSTTD and LSTAN was highest in the highest and lowest temperature prediction models.
Due to the nature of optical satellites, contamination of images caused by clouds increases the frequency of missing values, and especially, it is believed that the effects of rain clouds in summer and snow clouds in winter have affected the modeling accuracy.
Fig 11 – SAT here are interpolated – why? I would suggest showing it only at the weather stations or also only at the weather stations. What is the benefit from your model to a simple interpolation? Also would like to see a map of LST.
: A simple comparison of the predicted and observed values at each weather station is possible, but the reason for interpolating the SAT is:
1. Confirmation of areas where heat waves and cold waves occur through interpolated SAT (assuming true values)
2. Comparison with predicted SAT distribution

Reviewer 3 Report
The purpose of this study is to analyze the correlation between surface air temperature (SAT) and land surface temperature (LST) based on land use when heat and cold waves occur and to predict the distribution of SAT using the long short-term memory (LSTM) of TensorFlow. The study is well written and brings important contributions to remote sensing.
I would just like to see the citation of more works on the topic, both in the introduction and in the discussion. I suggest separating the results of the discussion.
I congratulate the authors.
Author Response
The purpose of this study is to analyze the correlation between surface air temperature (SAT) and land surface temperature (LST) based on land use when heat and cold waves occur and to predict the distribution of SAT using the long short-term memory (LSTM) of TensorFlow. The study is well written and brings important contributions to remote sensing.
: The manuscript has been modified to reflect the points as you pointed out. Thank you for your advice on making it a better paper.
I would just like to see the citation of more works on the topic, both in the introduction and in the discussion. I suggest separating the results of the discussion.
: (Line 403-504 and Reference number 17, 18, 39, and 51 to 55) The latest references have been added and the results and discussion were separated, with the additional analysis are summarized in the discussion.

Reviewer 4 Report
As shown in Section 3.1, the correlation between surface temperature and air temperature during cold and heat waves periods is not high compared to the whole period. Because cold and heat waves are caused by atmospheric pressure distribution, westerly winds, seasonal winds, and Fern phenomenon, etc. Therefore, it is unsuitable to predict air temperature only from surface temperature for cold and heat waves periods. Please explain why you think it is appropriate to apply the method of estimating air temperature only from surface temperature to these periods.
In Figure 8, the reason why air temperature was low on January 27, 30, February 7 and high on July 20, August 1, 13 must be analyzed. However, it cannot be explained only by surface temperature.
Author Response
As shown in Section 3.1, the correlation between surface temperature and air temperature during cold and heat waves periods is not high compared to the whole period. Because cold and heat waves are caused by atmospheric pressure distribution, westerly winds, seasonal winds, and Fern phenomenon, etc. Therefore, it is unsuitable to predict air temperature only from surface temperature for cold and heat waves periods. Please explain why you think it is appropriate to apply the method of estimating air temperature only from surface temperature to these periods.
: (Line 455-468) As you mentioned, heat and cold waves can be caused by various factors and there may be obvious limitations to predicting the SAT solely from the relationship between SAT and LST. However, it is difficult to deny that the LST and SAT may have a certain correlation. The prediction of SAT using existing models such as GDAPS, RDAPS, or other prediction models can be detailed and precise, but the wider the spatiotemporal range to be predicted, the more input data and computational resources are required. Therefore, this study aims to increase efficiency by simplifying the prediction of SAT which can be complex and to evaluate the prediction performance. Accordingly, we analyzed the characteristics of SAT and LST data based on land use and predicted the SAT using this relationship through TensorFlow. Also, this content has been added to the discussion.
In Figure 8, the reason why air temperature was low on January 27, 30, February 7 and high on July 20, August 1, 13 must be analyzed. However, it cannot be explained only by surface temperature.
: (Line 279-282 and 287-290) We have added an explanation of the reason why the SAT is high and low on those days as a point of view of meteorology.

Round 2
Reviewer 4 Report
The authors agreed with the reviewer's point that air temperature during the heat wave and cold wave cannot be predicted only from the surface temperature. Therefore, the method presented in this paper is not suitable for analysis of air temperature distribution during heat wave and cold wave.
Author Response
The authors agreed with the reviewer's point that air temperature during the heat wave and cold wave cannot be predicted only from the surface temperature. Therefore, the method presented in this paper is not suitable for analysis of air temperature distribution during heat wave and cold wave.
: (Line 403-421) As you mentioned, heat and cold waves can be caused by various factors and there may be obvious limitations to predicting the SAT solely from the relationship between SAT and LST. This study aims to increase efficiency by simplifying the prediction of SAT which can be complex and to evaluate the prediction performance. Accordingly, we analyzed the characteristics of SAT and LST data based on land use and predicted the SAT using this relationship through TensorFlow. As a result of additional verification for the heat wave period in 2019, the overall result was improved compared to 2018, and the possibility of SAT prediction using only LST was shown.
